# Scalable Feature Extraction and Tracking (SCAFET): A general framework for feature extraction from large climate datasets

Arjun Babu Nellikkattil[1,2], Danielle Lemmon[1,3,4], Travis Allen O'Brien[5,6], June-Yi Lee[1,2,7], and Jung-Eun Chu[8]

[1]Center for Climate Physics, Institute for Basic Science (IBS), Busan, South Korea, 46241
[2]Department of Climate System, Pusan National University, Busan, Republic of Korea, 46241
[3]Pusan National University, Busan, Rep. of Korea, 46241
[4]The American Association for the Advancement of Science, Science and Technology Policy Fellowship Program, Washington D.C., United States
[5]Department of Earth and Atmospheric Sciences, Indiana University Bloomington, Indiana, USA 47403
[6]Climate and Ecosystem Sciences Division, Lawrence Berkeley National Lab, Berkeley, USA 95720
[7]Research Center for Climate Sciences, Pusan National University, Busan, Republic of Korea, 46241
[8]Low-Carbon and Climate Impact Research Centre, School of Energy and Environment, City University of Hong Kong, Hong Kong, China

**Correspondence:** Arjun Babu Nellikkattil (arjunbabun@pusan.ac.kr)

**Abstract.** This study describes a generalized computational mathematical framework, Scalable Feature Extraction and Tracking (SCAFET) to extract and track features from large climate datasets. SCAFET utilizes novel shape-based metrics that can identify and compare features from different mean states, datasets, and between distinct regions. Features of interest such as atmospheric rivers, tropical and extratropical cyclones, jet streams, etc. are extracted by segmenting the data based on a scale-independent bounded variable called shape index (SI). SI gives a quantitative measurement of the local geometric shape of the field with respect to its surroundings. Compared with other widely used frameworks in feature detection, SCAFET does not use *a posteriori* assumptions about the climate model or mean state to extract features of interest and levelize comparison between different models and scenarios. To demonstrate the capabilities of the method, we illustrate the detection of atmospheric rivers, tropical and extratropical cyclones, sea surface temperature fronts, and jet streams. Cyclones and atmospheric rivers are extracted to show how the algorithm identifies and tracks both nodes and areas from climate datasets. The extraction of sea surface temperature fronts exemplifies how SCAFET effectively handles curvilinear grids. Lastly, jet streams are extracted to demonstrate how the algorithm can also detect three-dimensional features. As a generalized framework, SCAFET can be implemented to extract and track many weather and climate features across scales, grids, and dimensions.

## 1 Introduction

The amount of climate data is growing exponentially owing to rapid expansions in both observational capabilities and computational power, driven in particular by the precision and insights offered by higher resolution models (Overpeck et al., 2011; Balaji et al., 2018). Frontier research like global cloud-resolving and large ensemble simulations leads not just to increased volume but also to inflated velocity, variety, veracity, and value (5Vs) (Marr, 2015; Guo, 2017; van Genderen et al., 2019) of

climate data. This makes the detection and comparative analysis of important atmospheric and oceanic features, such as atmospheric rivers (ARs), tropical and extratropical cyclones, sea surface temperature fronts (SSTFs), and jet streams, an onerous task. Although these climate phenomena influence regional and global weather and climate with immense societal, economic, and ecological impacts, the amount of data representing these events and features is a small percentage of the whole simulation. Feature extraction considerably reduces the amount of data that needs to be stored, improving computational efficiency in analyzing these features (Yang et al., 2016). Moreover, the mean, variability, and characteristics of features can be compared to observational data sets as a measure of bias within model simulations, improving our understanding about the causal differences between observations and models (Sellars et al., 2013). Thus, efficient and reliable feature extraction is vital to climate data processing, analysis, and model development.

Despite the importance of feature extraction in climate data analysis and model development, there is little consensus on standard best practices for feature extraction. The simplest method for extracting a feature is to use a physical threshold or its derivative for some climate variable (SST, precipitation, wind speed, humidity, etc.), or a combination thereof, to identify ARs, fronts, jet streams, or tropical and extratropical cyclones (Bengtsson et al., 1982, 1995; Vitart et al., 1997; Hewson, 1998; Koch et al., 2006; Strong and Davis, 2007; Rutz et al., 2014; Guan and Waliser, 2015). The limitations and discrepancies in these methods arise from the somewhat arbitrary choice of physical thresholds in relation to the spatiotemporal distributions of the climate variables. In other words, many studies choose a physical threshold that is not theoretically defined but rather a function of the location, time span, and dataset used. Validation can then unfortunately come down to the intelligent but subjective human eye, or in other words tuning an absolute or relative threshold until it appears to have captured all the features of interest while leaving out the background noise (Zarzycki and Ullrich, 2017; Vishnu et al., 2020).

Choosing an absolute threshold from climate variables for feature extraction that applies to different climate models and spans multiple mean states and model scenarios is not straightforward. Thresholds are often applied to climate variables or derivatives in which the features are most visible, such as relative vorticity (RV) and sea level pressure anomalies for tropical cyclones (e.g., Vitart et al., 1997), integrated water vapor transport (IVT) for ARs (e.g., Guan and Waliser, 2015), or the first derivative of sea surface temperature (SST) for SST fronts (Castelao et al., 2006). Thresholds are often either empirically derived from observational studies or calculated from a model-specific distribution, though even within the same dataset a particular choice of threshold may be suitable for one region but not for another, given varying regional characteristics and topography. In the case where the feature extraction threshold is an *a posteriori* assumption of the data set used, one must preprocess large, representative datasets just to calculate reasonable thresholds. While some detection methods have done well to streamline their algorithms to reduce total runtime, the process of posterior threshold calculation for higher resolutions and large ensemble datasets inherently becomes increasingly less efficient, highlighting the need to develop feature extraction methods that do not use posterior assumptions.

Aside from the sensitivity of feature detection to inter-model and inter-simulation differences, feature detection is further complicated when trying to detect and compare features between present and future climate change scenarios as the underlying spatiotemporal climate variable distributions change under global warming. Feature detection must be reconsidered when applied to variables with significant and/or non-linear changes in their means and extremes in response to external forcings

such as doubling or quadrupling carbon dioxide concentrations. It should be emphasized that applying different arbitrary

thresholds can and does lead to contradictory conclusions regarding the response of these features to greenhouse gas warming (Horn et al., 2014; Zhao, 2020; O'Brien et al., 2022; Nellikkattil et al., 2023). To counter these uncertainties, methods based on topology, machine learning, ridge extraction, edge detection, and various other image-processing techniques have been proposed over the years (Dixon and Wiener, 1993; Post et al., 2003; Molnos et al., 2017; Biard and Kunkel, 2019; Xu et al., 2020). While these methods offer an alternative for the extraction of features in datasets spanning different mean states, many

of these methods were developed for detecting specific rather than general features.

The need for a general framework for extracting and tracking features from large climate datasets has been raised in various climate science communities for the last several decades. In a pioneer study, Hodges (1994) developed a general framework for extracting and tracking features from meteorological datasets in three steps: segmentation, filtering, and tracking. In the segmentation step, the field is split into distinct regions by applying a threshold and defining each of the connected regions as

an object. Segmented regions are then filtered based on the characteristics of each object, and feature nodes are defined for the remaining objects. Finally, the feature nodes are tracked over time to produce the final output for further analysis. This framework was further developed for cyclones, storm tracks, convective systems, ocean eddies, monsoon depressions, etc., (Hodges, 1995; Hogg et al., 2005; Hodges et al., 2011; Burston et al., 2014; Hurley and Boos, 2014; Pinheiro et al., 2016; Priestley et al., 2020; Torres-Alavez et al., 2021; Karmakar et al., 2021). However, it is limited to the detection of points of

local maxima in two-dimensional scalar fields, which do not always fully characterize various features.

In 2012 a team from the Lawrence Berkeley National Laboratory developed the Toolkit for Extreme Climate Analysis (TECA), integrating pre-existing, physical threshold-dependent detection methods and algorithms into a comprehensive software package that was parallelized to make the algorithms more suitable for large datasets (Prabhat et al., 2012). In a more recent effort, a team led by Paul Ullrich at the University of California-Davis created TempestExtremes (Ullrich and Zarzycki,

2017; Ullrich et al., 2021), another computationally efficient algorithm package that uses C++ and several core functions to detect a variety of features. These functions are being actively developed for extraction, characterization, and uncertainty quantification of weather extremes. Both TECA and TempestExtremes have been widely implemented by the climate community and have been monumental in advancing scientific understanding of meso- and synoptic-scale processes and their connections to long-term climate variability.

In this study, we present a novel method called Scalable Feature Extraction and Tracking (SCAFET), which serves as a versatile and general framework for detecting and tracking features of various shapes and intensities across scales, grid types, and dimensions. Simply put, SCAFET uses the curvature measurements of a given scalar field to identify distinct emergent shapes corresponding to features of interest. The local shape calculation is finite, bounded, and scale-independent, and it can be tuned depending on the specified feature of interest. Unlike traditional methods that rely on physical thresholds often derived

from data-specific, posterior conditions, this method relies on shape-based thresholds. As such, it separates the feature detection process from inter- and intra-model variation, making it less sensitive to these differences. Furthermore, this approach allows for the complete parallelization of feature extraction along the time dimension since the detection operates independently of time. Time-independent feature extraction offers two key advantages. Firstly, it has the potential to boost computational efficiency by

enabling data pre-processing such as smoothing to occur in parallel, rather than requiring a single pre-processing step before
feature extraction. Secondly, it holds the promise of being developed and implemented for real-time feature extraction during
critical events like hurricanes and tornadoes. Importantly, the code for this framework is fully open-source and written in Python
in an easy-to-use package so that even individuals with beginner-level Python skills can readily implement the algorithm (see
https://github.com/nbarjun/SCAFET/blob/master/scafet_demo.ipynb for a simple working example).

The novelty of SCAFET compared to pre-existing methods lies in feature detection that does not use *a posteriori* assumptions
and is based on the overall "shape" of a climate variable field, rather than arbitrary thresholding of that field or derivative. The
core methodology for the detection of any feature is the same and can be tuned using just two variables, one for the spatial
scale and the other for the shape of the features one is looking for. For example, between the two variables, one can tune the
difference between a long filament-shaped atmospheric river and a shorter round-shaped cyclone. The algorithm applies to
both rectilinear and curvilinear grids and can also be extended to detect three-dimensional (3D) features. Even in the context
of recent advancements in feature extraction such as Tempest Extremes and TECA, SCAFET is a comprehensive, efficient,
and easily implementable framework that aims to upgrade the feature extraction process with a novel shape-based approach
that does not rely on iterative posterior conditions and could prove to be a robust method for detecting a diverse set of features
under different mean climate states. Further discussion on the differences between SCAFET and other detection algorithms
can be found in Appendix A.

The paper is organized as follows, section 2 introduces the fundamentals of SCAFET and how it is implemented in a two-
dimensional (2D) field. Section 3 presents three specific use cases of SCAFET, demonstrating its capabilities in detecting
various climate features across different grid types. Extraction of 3D features using jet steams as an example will be discussed
in section 4. Though the application of SCAFET is not limited to the features described here, this study focuses on atmospheric
rivers, cyclones, SST fronts, and jet streams as these examples cover a broad range of phenomena, providing users with insights
on how to adapt SCAFET to their specific use cases and requirements.

## 2 Description of Scalable Feature Extraction and Tracking

SCAFET adopts the same three-step approach as outlined by Hodges (1994) -Segmentation (yellow boxes in Figure 1) , Fil-
tering (orange boxes in Figure 1), and Tracking (green boxes in Figure 1). However, before commencing these steps, SCAFET
requires initialization with essential information describing the datasets and the specific feature to be extracted (as indicated by
blue boxes in Figure 1). The key inputs for this initialization include the following:

– **Primary field** ($\phi_p$): This is a gridded dataset in which the target feature is most easily distinguishable. For instance,
cyclones are readily identified using the RV field, ARs emerge from IVT, and SSTFs are distinguished using the SST
gradient. Optionally, one or more secondary fields can be used to further constrain the detected features.

– **Grid Properties**: Information on the primary field's grid including grid cell area/volume, grid distance, and coastlines
are required for calculating derivatives of the basic field and identifying landfalling locations.

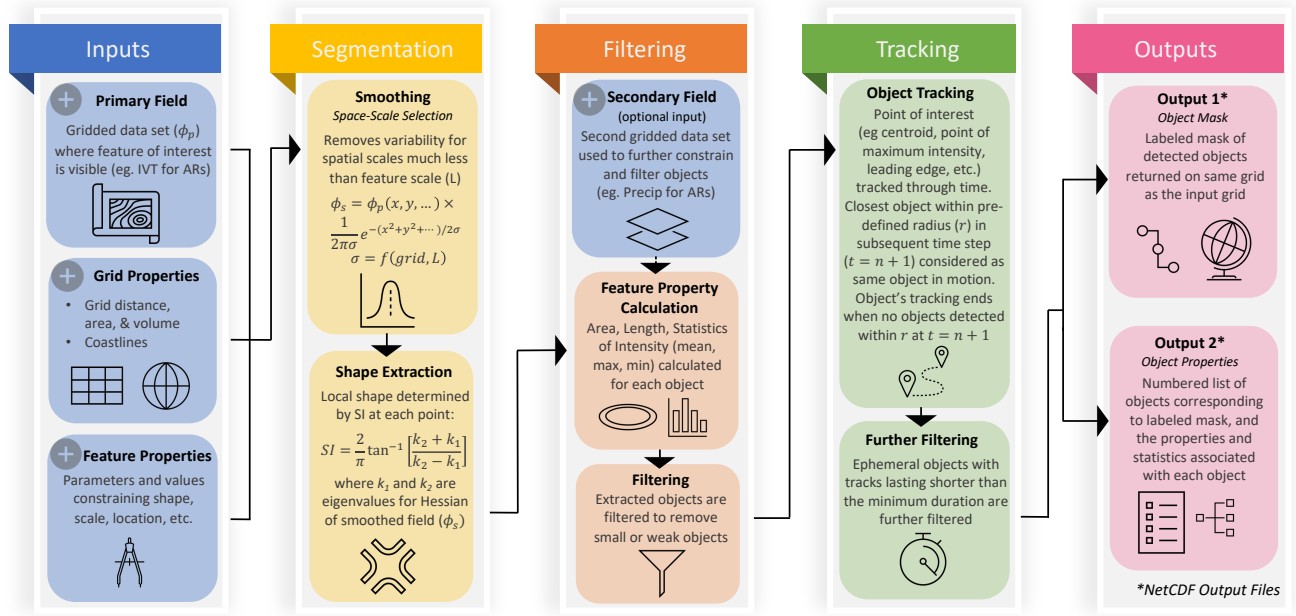

**Figure 1.** Overall schematic of SCAFET workflow and components. Inputs to the algorithm are depicted in blue, while the algorithm's outputs are shown in pink boxes. Processes related to the segmentation step are highlighted in yellow boxes, whereas the orange boxes represent the filtering processes. The tracking step is denoted by green boxes. Arrows on the periphery of the boxes illustrate the flow of the algorithm. Each section is elaborated upon in detail within the text.

- **Feature Properties**: The algorithm requires information on the properties of the target feature. This includes estimated spatial scale, shape, eccentricity (for 2D features only), minimum length, minimum area, minimum volume (for 3D feature only), minimum duration, and maximum distance per time step.

In the SCAFET scheme, segmentation, filtering, and tracking are developed and coded as separate Python libraries. This design allows users to substitute any of these components with their own methods while still being able to execute the algorithm. Once all three steps have been executed, the algorithm yields two outputs: one provides information about the properties of the detected objects, and the other produces a labelled mask highlighting the feature of interest on the input grid (pink boxes in Figure 1).

## 2.1 Segmentation

The core operation for the feature extraction involves categorizing points within a scalar field into one of five shapes. This categorization is achieved using curvature measurements obtained from the eigenvalues of the Hessian of the basic field. These

five selected shapes (see Figure 2) are an abridged version of the shapes described in previous studies (Koenderink and van Doorn, 1992). Depending on the specific feature of interest, one or more shapes are extracted from the primary field. The segmentation process starts with scale-space selection of the field to remove smaller scales of variability that are background noise compared to the feature of interest. Lastly, the algorithm calculates SI to estimate the local geometric shape at each point.

### 2.1.1 Scale-space Selection

Scale-space selection is a widely used technique in image processing, signal processing, and computer vision (Lindeberg, 2014). In our current study, scale-space selection involves applying a Gaussian smoothing kernel to suppress variability smaller than the chosen smoothing scale ($\sigma$) (see https://unidata.github.io/MetPy/latest/api/generated/metpy.calc.smooth_gaussian.html for implementation of Gaussian smoothing). Mathematically, scale-space selection is performed by convolving the primary field ($\phi_p$) with a Gaussian function, expressed as follows:

$$\phi_s(x,y,\dots) = \phi_p(x,y,\dots) * \frac{1}{2\pi\sigma} e^{-(x^2+y^2+\dots)/2\sigma^2} \tag{1}$$

In the context of the meso-synoptic scale processes examined in this study, scale-space selection filters out smaller micro-scale features to isolate features like cyclonic vortexes or atmospheric rivers. Notably, this function can be adjusted to the spatial scale of interest and could also be used to filter out synoptic-scale features in isolating micro- and meso-scale processes. In climate datasets, grid spacing is not always uniform. To account for that, we adapt the above equation to be "grid-aware". The input for the smoothing scale is provided in kilometers, and based on this input, we calculate the value of $\sigma$ while considering the grid size. Notably, the value of $\sigma$ remains constant when smoothing is applied along each longitude, but it varies along each circle of latitude. For future studies, researchers may explore other, more advanced scale-space selection methods to further refine their analyses.

### 2.1.2 Local Shape Extraction

The local geometric shape of the field, $\phi_s$ is calculated as a function of the eigenvalues ($k_1$ and $k_2$) of the Hessian of the magnitude of the field ($|\phi_s|$), where the Hessian is given by,

$$\mathcal{H}\left(|\phi_s|\right) = \begin{bmatrix} \frac{\partial^2 |\phi_s|}{\partial x^2} & \frac{\partial^2 |\phi_s|}{\partial x \partial y} \\ \frac{\partial^2 |\phi_s|}{\partial y \partial x} & \frac{\partial^2 |\phi_s|}{\partial y^2} \end{bmatrix} \tag{2}$$

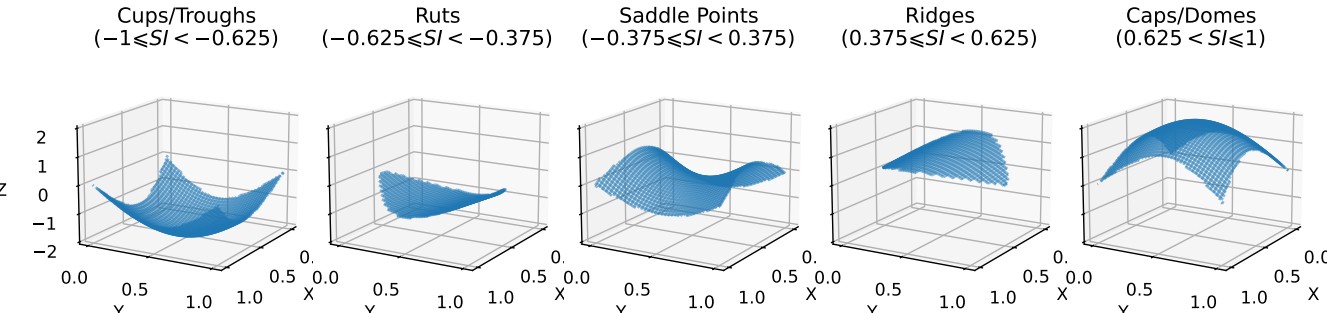

**Figure 2.** Selected shapes used in this study and the values of the shape index associated with each of them. The X and Y axis are a set of general axis while $Z(X,Y) = \sin(2X) + \cos(2Y)$. Regions within $Z(X,Y)$ satisfying conditions for different shapes are isolated to show the geometry associated with them.

In the context of simple differential geometry, we can determine whether a point is a local maximum or a local minimum based on the eigenvalues $k_1$ and $k_2$. Specifically, if $k_2 \leq k_1 < 0$ then the point under consideration is a local maximum, whereas if $k_1 \geq k_2 > 0$, the point is a local minimum. The criterion is primarily applicable to nodal features such as tropical cyclones or monsoon depressions. To expand our ability to identify a range of features, we use shape index (SI) (Koenderink and van Doorn, 1992), a quantitative measure of the local shape of the field defined as,

$$SI(k_1, k_2) = \frac{2}{\pi} \tan^{-1} \left[ \frac{k_2 + k_1}{k_2 - k_1} \right] \tag{3}$$

Where $k_1$ and $k_2$ are the two eigenvalues, satisfying $k_1 \geq k_2$, for the Hessian matrix. It is important to clarify that in the original work by (Koenderink and van Doorn, 1992), the principal curvatures, not the eigenvalues of the field, are utilized to calculate the SI. However, the disparity between SI calculated using principal curvatures and SI derived from eigenvalues is exceedingly minimal in climate data analysis. The SI is used to categorize the primary field into distinct shapes (see Figure 2). The choice of SI values is contingent upon the specific type of feature to be extracted. For example, we select caps and domes when extracting features such as atmospheric depressions or cyclones, whereas ridges, caps, and domes are chosen when targeting features like ARs and fronts.

SI is designed to be a bounded value (range -1 to 1) independent of the magnitude of the field (Figure 3). In simple terms, SI provides a continuous and quantitative measurement of the geometric shape of the field with respect to its immediate background field. This concept is similar to how a climate scientist's trained eye identifies features based on differences in color or value contrast, though SI is arguably a more objective and precise measure of geometric shape. These characteristics make SI particularly well-suited for feature extraction from datasets with varying mean states, in contrast to traditional physical threshold-based methods. In addition to the two eigenvalues, the shape extraction provides us with corresponding eigenvectors. The eigenvector for $k_1$ points perpendicular to the local ridge direction while that of $k_2$ is parallel to it. This allows us to impose further constraints, such as the coherence of transport or flow with respect to the local ridge when $\phi_s$ is a vector field. This capability is aptly demonstrated in the context of AR detection, as discussed in subsection 3.1.

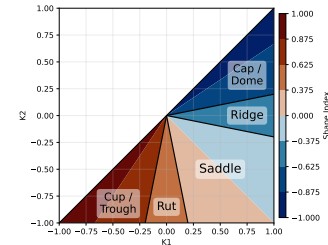

**Figure 3.** Sensitivity of the shape index (SI) to eigenvalues $k_1$ and $k_2$. The X and Y axes represent values of the two eigenvalues used for calculating the shape index while the color indicates the value of the shape index. Shapes corresponding to SI regimes are labelled. Shapes corresponding to SI regimes are labelled.

## 2.2 Filtering

Once the target features are extracted, properties like area, location, mean, minimum, and maximum values of different properties are calculated for each of the objects. A series of filtering is carried out to remove objects that do not satisfy certain conditions regarding (a) grid properties like area, length, region masks, etc. (b) primary field properties like magnitude and direction, and (c) constraints from the secondary field(s). The primary aim of the filtering process is to remove small, weak, or ephemeral objects.

## 2.3 Tracking

The properties extracted for each object include key positional details, such as its centroid, and endpoints, as well as the locations of maximum and minimum intensity of input field within each object. To follow objects through time, one of these positional attributes is tracked. In the present study, we employ a straightforward tracking method. For each object at time step $n$, we identify the closest object to it at time $n + 1$. If this identified object is closer than a predefined radius $r$, we consider it to be the same object in motion. The radius $r$ is defined in kilometers based on the maximum translation speed of the object and the temporal frequency of the input data. At this stage, it is possible to filter out short-lived features as needed. While this uncomplicated tracking approach may not be suitable for micro-scale processes, it can be adapted to incorporate greater complexity if necessary.

## 3  Application to 2D Features

In this section, we showcase how SCAFET is employed to detect cyclonic vortices, ARs, and SSTFs from various climate datasets. These examples serve to illustrate the versatility of SCAFET as applied to different types of features and grids, though all the examples in this study follow the same general process shown in Figure 1. Each subsection has a table of parameters detailing the properties of the desired feature. The properties include the feature's typical spatial scales, shape index (SI)

regime, minimum length, minimum area, object eccentricity, and minimum duration of its track. To determine the quantitative values for these properties, we refer to a consensus among previous studies, which are cited within each section. A detailed examination of the sensitivity of these parameters in relation to the detected features, using AR detection as an example, can be found in Supplementary Section 1. In addition to the results discussed in the following sections, supplementary videos are also included for each of the features. The primary objective of this work is to demonstrate SCAFET's capability to detect a variety of features. Consequently, we present results for the long-term climatology of each of the features, enabling a comparison with other published detection algorithms.

## 3.1 Atmospheric Rivers

According to the American Meteorology Society's glossary of meteorology, ARs are "long, narrow, and transient corridors of strong horizontal water vapor transport that are typically associated with a low-level jet stream ahead of the cold front of an extratropical cyclone" (Ralph et al., 2018). A substantial portion of the precipitation and water vapor transport in midlatitude regions is concentrated within ARs (Guan and Waliser, 2015). These atmospheric phenomena play a significant role in midlatitude hydrology, contributing to more than 50% of the extreme precipitation and wind events in the region (Waliser and Guan, 2017; Nash et al., 2018). The ability to accurately detect, forecast, and project future ARs is of utmost importance for both extreme weather preparedness as well as for water resource management in basins worldwide.

The ambiguity in AR projections and AR detection tools (ARDTs) stems from the lack of a clear quantitative definition of ARs in strength, length, narrowness, and other such parameters used in detection. In comparison with other criteria, the choice of threshold for AR strength has a significant effect on the inferences drawn between the detection schemes (Zhao, 2020; O'Brien et al., 2022; Nellikkattil et al., 2023). Many ARDTs determine this threshold empirically from the dataset itself, which renders them sensitive to spatiotemporal variations and changes in mean-state conditions (Shields et al., 2018). SCAFET defines ARs as long (length > 2000 km), narrow (eccentricity > 0.75) regions of strong water vapor transport (SI > 0.375), and significant precipitation (minimum AR precipitation > $1\,\mathrm{mm\,day}^{-1}$) (see Table 1 for complete details). The sensitivity of these parameters in AR detection to the characteristics of detected ARs is discussed in Supplementary Section 1. This approach reduces the sensitivity of AR characteristics to arbitrary strength thresholds, making it easier to compare ARs across different mean state conditions.

To illustrate how SCAFET identifies ARs, we utilized daily mean data from the European Centre for Medium-Range Weather Forecasts (ECMWF) Reanalysis Version 5 (ERA5; Hersbach et al. (2020)) for the period 2000 to 2019. The key fields of interest included the daily mean integrated water vapor transport (IVT) as the primary field and the daily mean total precipitation as the secondary variable. All the datasets employed share a spatial resolution of $0.25° \times 0.25°$. The vector field, IVT is calculated as,

$$IVTx = -\frac{1}{g} \int\limits_{1000hPa}^{300hPa} q.\mathbf{U}dp \qquad (4)$$

$$IVTy = -\frac{1}{g} \int\limits_{1000hPa}^{300hPa} q.\mathbf{V} dp \tag{5}$$

$$|IVT| = \sqrt{IVTx^2 + IVTy^2} \tag{6}$$

To detect AR-like structures, SCAFET employs a search for specific shapes, such as ridges, caps, and domes (see Figure 2). Following the process outlined in Figure 1, SI is calculated after applying a grid-aware smoothing technique that suppresses variability smaller than 1000 km (Figure 4(a)). Once SI is calculated for |IVT| (Figure 4(b)), regions where SI > 0.375 are passed on to the next stage for filtering. To maximally utilize the vector qualities of the primary field, we ensure that the local transport direction (arrows in Figure 4(a)) and local ridge direction (arrows in Figure 4(b)) do not deviate by more than $45°$. The local ridge direction is identified as the eigenvector corresponding to the smallest eigenvalue ($k_2$). Filtering based on the grid properties removes candidates that are too small (length < 2000 km and area < $2 \times 10^6 \mathrm{km}^2$), or too wide (eccentricity $\leq 0.75$). To eliminate AR-like objects with low strength (precipitation < $1 \mathrm{\,mm\,day}^{-1}$) we constrain our results with the secondary field, total precipitation, within the object's area. The use of precipitation as a strength indicator is relevant given its significant socio-economic impact. In line with other ARDTs, we impose a regional mask to filter out AR-like structures along the equatorial belt. All these steps can be applied in parallel along the time axis, and at each time step AR-like structures similar to those shown in Figure 4(c) are identified. Once all ARs are detected, the tracking algorithm is applied to the daily data to filter out ARs that last less than one day. Tracking is performed based on the centroid of each identified object. The closest objects within a distance of 4000 km between two consecutive time steps are considered the same object evolving over time (Figure 4(d)). The annual mean frequency of the detected AR objects and their seasonality are shown in Figure 4(e), (f), and (g). SCAFET's identification of ARs is consistent with other ARDTs, both in terms of detecting single events and determining their mean climatology, as further detailed in the Supplementary Section 2 (see also Lora et al. (2020)).

### 3.2 Tropical and Extratropical Cyclones

In the scientific literature, cyclones are generally described as large weather systems ranging from 500–4000 km in size, characterized by strong cyclonic circulation, low pressure at their center, and exceptionally high winds around it (Emanuel, 2003; Schultz et al., 2019; Encyclopaedia, 2022). The dynamics and characteristics of cyclones can vary depending on factors such as their genesis location and translation speeds. For instance, cyclones generated near the equator, commonly referred to as tropical cyclones, are typically smaller in size compared to those formed in midlatitudes, known as extratropical cyclones. Regardless of their origin, cyclones have the potential to unleash intense rainfall, powerful winds along their path, and can lead to flooding, landslides, and severe damage to coastal infrastructure when they make landfall (Knutson et al., 2010; Mendelsohn et al., 2012; Ranson et al., 2014). Moreover, the impact of cyclones is becoming a subject of heightened public concern due to rising sea levels and the potential for increased cyclone intensity in response to global warming. Thus, the identification and future projection of cyclones are a subject of growing attention and importance for the climate community (Woodruff et al., 2013).

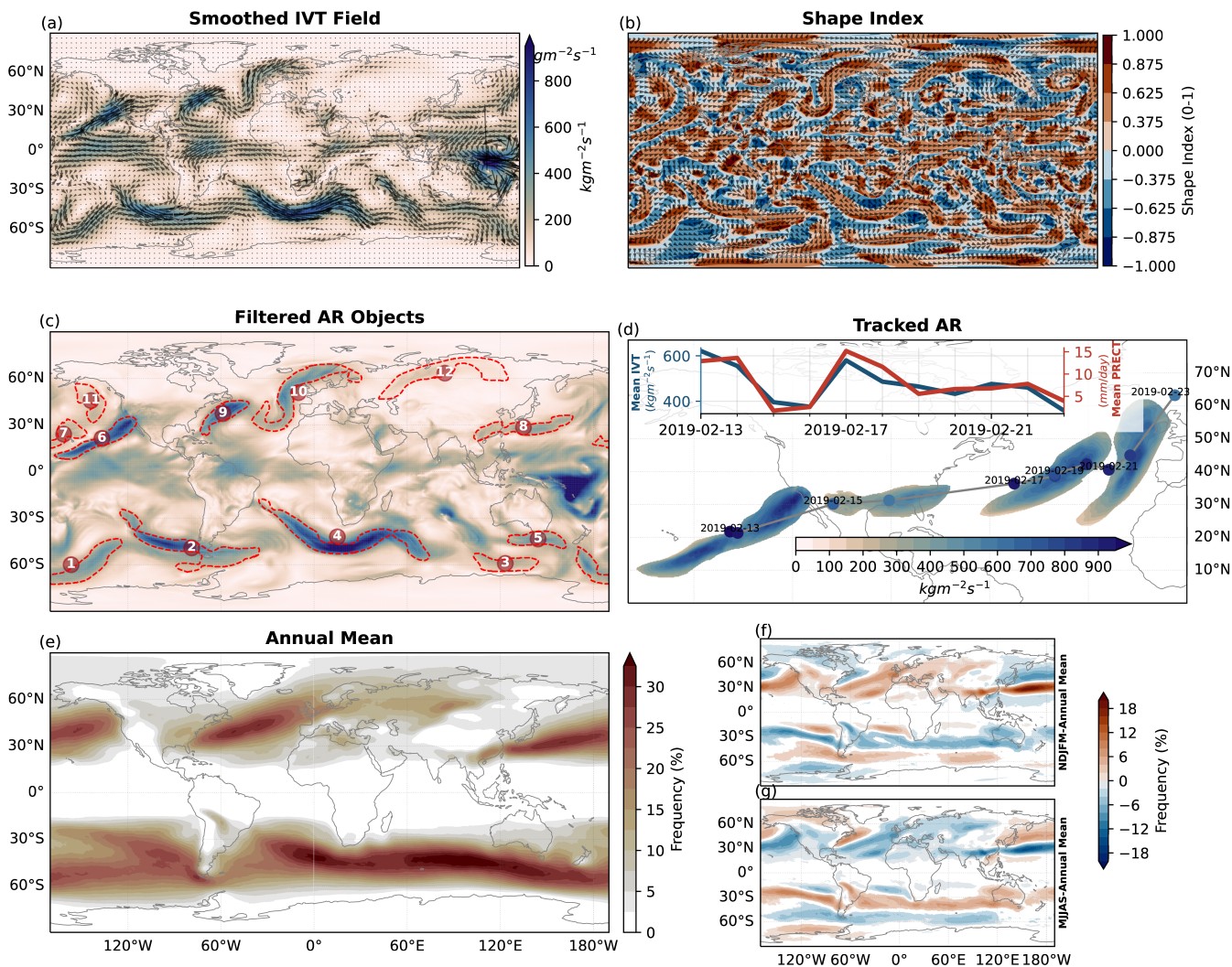

**Figure 4.** Major steps in the detection and tracking of Atmospheric Rivers. (a) Smoothed primary field of vertically integrated water vapor transport (IVT). Smoothing removes variability smaller than 1000 km from the IVT. The arrows in (a) represent the direction of unsmoothed IVT. (b) Magnitude (shading) of the shape index (SI) and direction of the local ridge (arrows) direction calculated from smoothed IVT. (c) Labeled AR objects after filtering out weak, small, and ephemeral candidates. (d) Example of tracked AR centroids and marked time, inlay shows object's area mean IVT over time. (e) AR annual mean frequency for the period 2000 to 2019. (f-g) AR Frequency anomaly relative to the annual mean for (f) November to March, and (g) May to September.

| | No. | Property | Value | Unit |
|---|---|---|---|---|
| **Segmentation** | 1 | Smooth Scale | 2000 | km |
| | 2 | Angle Coherence | 45 | degrees |
| | 3 | Selected Shape | (0.375,1.0] | - |
| **Filtering** | 1 | Minimum Length | 2000 | km |
| | 2 | Minimum Area | $2\times10^6$ | km$^2$ |
| | 3 | Eccentricity | [0.75, 1.0] | - |
| | 4 | Minimum Precipitation | 1 | mm day$^{-1}$ |
| | 5 | Latitude Mask | (-20, 20) | degrees |
| **Tracking** | 1 | Minimum Duration | 24 | hours |
| | 2 | Maximum Distance per Timestep | 4000 | km |

**Table 1.** The table presents the values for various parameters used in the detection of ARs using SCAFET. The rows for each step, including segmentation, filtering, and tracking are grouped together and labelled.

Once again, discrepancies among different detection algorithms can be attributed to varying choices of physical thresholds or constraints related to factors such as size, wind speeds, vorticity, or surface pressure anomalies. While most studies generally agree on the present and future characteristics of cyclones, resolving details such as the changes in genesis rate and durations is complicated by the uncertainties in the detection methods (Ulbrich et al., 2009; Neu et al., 2013; Horn et al., 2014; Walsh et al., 2015). In this study, SCAFET identifies cyclones as regions of strong local maxima of cyclonic circulation (SI > 0.625) and maximum wind speeds exceeding $10\,\mathrm{m\,s^{-1}}$. This definition enables the detection of robust cyclonic vorticities worldwide, including but not limited to tropical and extratropical cyclones. The primary field used for cyclone detection is the absolute value of cyclonic relative vorticity ($\zeta$) defined as,

$$\zeta = \nabla \times \mathbf{U} \tag{7}$$

Where $U$ is the 6-hourly wind speeds at 10 meters above the surface obtained fro1mmdaym the ERA5 reanalysis dataset with a spatial resolution of $0.25° \times 0.25°$ (Hersbach et al., 2020). The magnitude of wind speed at 10 meters is utilized as

| | No. | Property | Value | Unit |
|---|---|---|---|---|
| **Segmentation** | 1 | Smooth Scale | 1500 | km |
| | 2 | Selected Shape | (0.625,1.0] | - |
| **Filtering** | 1 | Minimum Length | 20 | km |
| | 2 | Minimum Area | $10^5$ | $km^2$ |
| | 3 | Eccentricity | [0.0,1.0] | - |
| | 4 | Minimum Vorticity | $10^{-6}$ | $s^{-1}$ |
| | 5 | Minimum Max. Windspeed | 10 | $ms^{-1}$ |
| **Tracking** | 1 | Minimum Duration | 48 | hours |
| | 2 | Maximum Distance per Timestep | 500 | km |
| | 3 | Net Minimum Displacement | 1000 | km |

**Table 2.** Same as in table Table 1, but for parameters and values relevant to detecting tropical and extratropical cyclones.

the secondary field to constrain detection. Additional cyclone-related variables such as surface pressure anomaly and potential temperature can also serve as secondary fields for the identification and classification of cyclones.

In contrast with ARs, the detection of cyclones relies on a scalar field, specifically in this case the cyclonic relative vorticity $|\zeta|$. First, the data is pre-processed with grid-aware Gaussian smoothing to suppress spatial variability smaller than 750 km (Figure 5(a)). The chosen smoothing scale allows us to identify both tropical and extratropical cyclones. Caps and dome shapes (SI > 0.625) are then identified within the smoothed $|\zeta|$ field as potential cyclones (Figure 5(b)). Subsequently, objects with an area less than $10^5 km^2$ and a diameter less than 20 km are filtered out. Once these spatial criteria are met, we can further refine our selection by excluding weak cyclonic vorticities $|\zeta| < 10^{-6} s^{-1}$ and slow maximum wind speed $< 10 \mathrm{\,m\,s^{-1}}$, resulting in the identification of robust cyclonic systems for a given time step (Figure 5(c)). Similar to the AR example, all the described steps can be parallelized along the time dimension. Once potential cyclones are identified, they are tracked using a methodology similar to the AR tracking algorithm. However, in this case, the radius for search is limited to 1000 km since we are using 6-hourly data, and the translation speeds of cyclones are notably slower than $150 \mathrm{\,km\,h^{-1}}$. A minimum duration of 48 hours and a minimum total displacement of 500 km is applied to distinguish moving cyclonic circulations from stationary ones. An example of a tracked cyclone, commonly known as "Dorian" (Avila et al., 2020) is compared with the observed

track from IBTrACS (Knapp et al., 2010, 2018) dataset (Figure 5(d)). In comparison to the observed track, SCAFET's track is much longer due to the more relaxed conditions applied to cyclonic vorticity and wind speed. Additionally, SCAFET does not

differentiate between tropical and extratropical cyclones, which can result in tracking the object throughout its transition from a tropical cyclone to a midlatitude storm. Despite this difference, the long-term averages for cyclone frequency and its seasonal variability calculated using SCAFET are comparable with other studies (e.g., Ullrich and Zarzycki, 2017). What sets SCAFET apart from other conventional cyclone detection algorithms is its approach to identifying cyclones not as point objects, but as encompassing surfaces around the point of maximum $|\zeta|$. This enables a more comprehensive analysis of cyclone properties,

including maximum and minimum values of wind speed and precipitation within the entire cyclone structure.

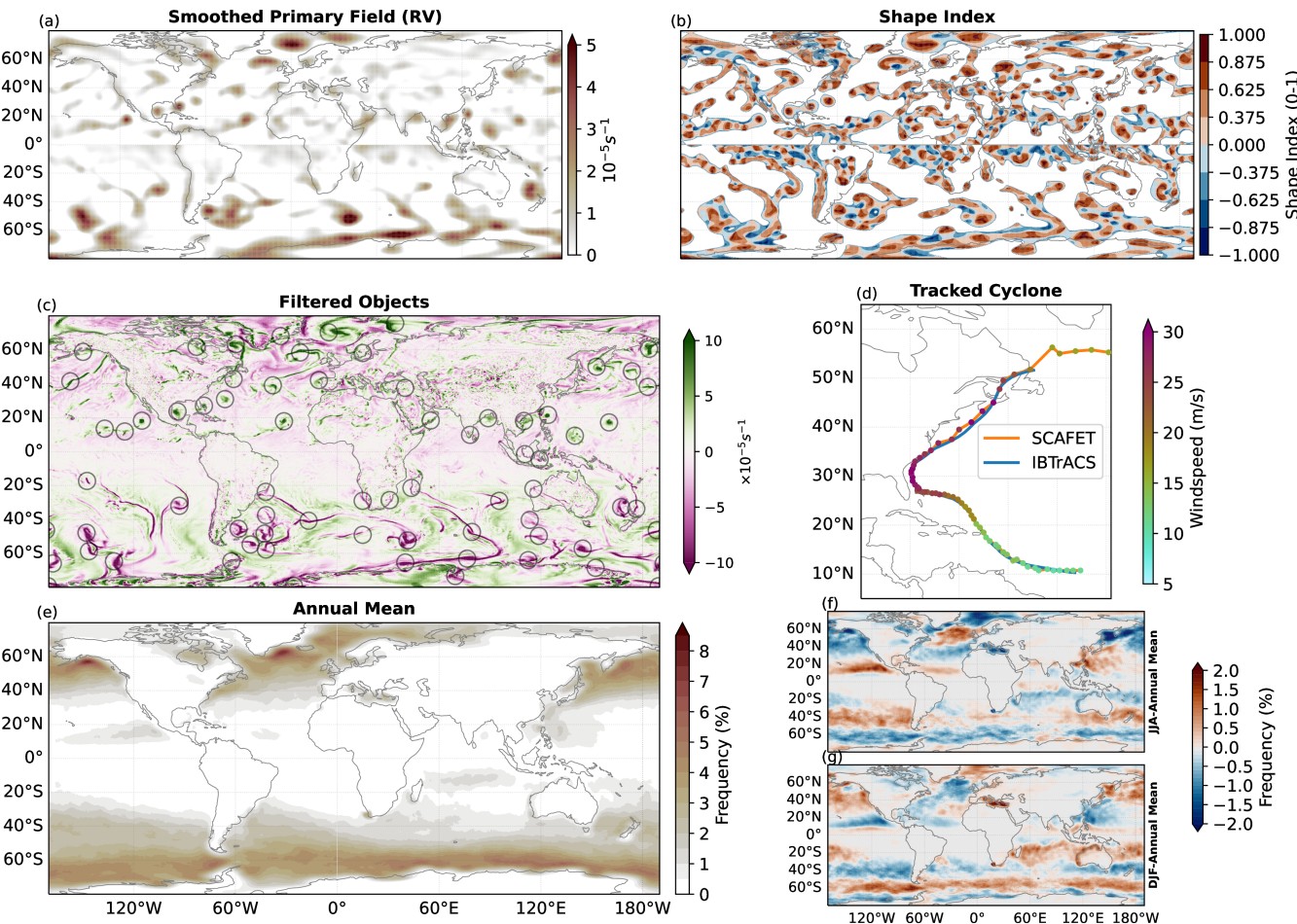

**Figure 5.** Major steps in the detection and tracking of cyclones. (a) Smoothed primary field of cyclonic relative vorticity ($|\zeta|$). The smoothing removes variability smaller than 750 kms from $|\zeta|$. (b) Magnitude of SI for the primary field. (c) Filtered cyclonic objects with the background color representing unsmoothed values of $\zeta$. (d) Track obtained for cyclone "Dorian" from SCAFET compared with the track from the IBTrACS dataset. (e) Annual mean frequency of cyclone occurrence for the period 2000 to 2020. (f-g) Anomalous cyclone frequencies relative to the annual mean for (f) JJA and (g) DJF.

### 3.3 Sea Surface Temperature Fronts

SST fronts are regions where different water masses come together. They are typically characterized by strong horizontal gradients in temperature, salinity, density, and other properties (Bowman, 1978; Legeckis, 1978; Fedorov, 1986; Yoder et al., 1994). Unlike the larger meso to synoptic scale features discussed in this study, frontal structures are often observed in much smaller spatiotemporal scales. Accurate identification of SSTFs is essential because these features are frequently associated with strong upwelling and high levels of biogeochemical productivity (Clayton et al., 2014, 2021; Nagai and Clayton, 2017). Additionally, the detection of SSTFs serves as an example of how SCAFET can be applied to identify features in curvilinear grids.

Many prior SSTF detection algorithms rely on edge detection techniques and the gradient of sea surface temperature and/or height to identify these structures (Canny, 1986; Castelao et al., 2006). In our approach, we utilize the magnitude of the daily mean SST horizontal gradient as the primary field for detecting SST fronts. The SST data is obtained from a fully coupled, ultra-high-resolution ($\approx 10\,km$) CESM v1.2.2 simulation of present-day mean climate (Small et al., 2014; Chu et al., 2020; Nellikkattil et al., 2023). The data is processed by SCAFET in the tripolar POP grid. To illustrate the detection process, the analysis focuses on the Kuroshio frontal and extension region for the last 10 years of the simulation.

The extraction of frontal structures using the selected shapes of ridges, caps, and domes is similar as in the detection of ARs. Prior to extraction, a spatial smoothing of approximately 30 km is applied. From the extracted SSTF candidates, objects with a mean SST gradient lower than $10^{-4}\mathrm{K\,m^{-1}}$ are removed. Circular (eccentricity < 0.5) and small (area < $1000\,\mathrm{km^2}$) objects are also filtered out. It is worth noting that, in contrast to AR detection, frontal structures are not tracked. The detected frontal frequency exhibits general patterns and seasonality consistent with findings in previous studies (Xi et al., 2022).

|  | No. | Property | Value | Unit |
|---|---|---|---|---|
| Segmentation | 1 | Smooth Scale | 30 | km |
|  | 2 | Selected Shape | (0.375, 1.0] | - |
| Filtering | 1 | Minimum Length | 500 | km |
|  | 2 | Minimum Area | $10^3$ | $\mathrm{km^2}$ |
|  | 3 | Eccentricity | (0.5,1.0] | - |
|  | 4 | Minimum SST Gradient | $10^{-4}$ | $\mathrm{K\,m^{-1}}$ |

**Table 3.** Same as in Table 1 but for parameters and values relevant to detecting Sea Surface Temperature Fronts (SSTFs).

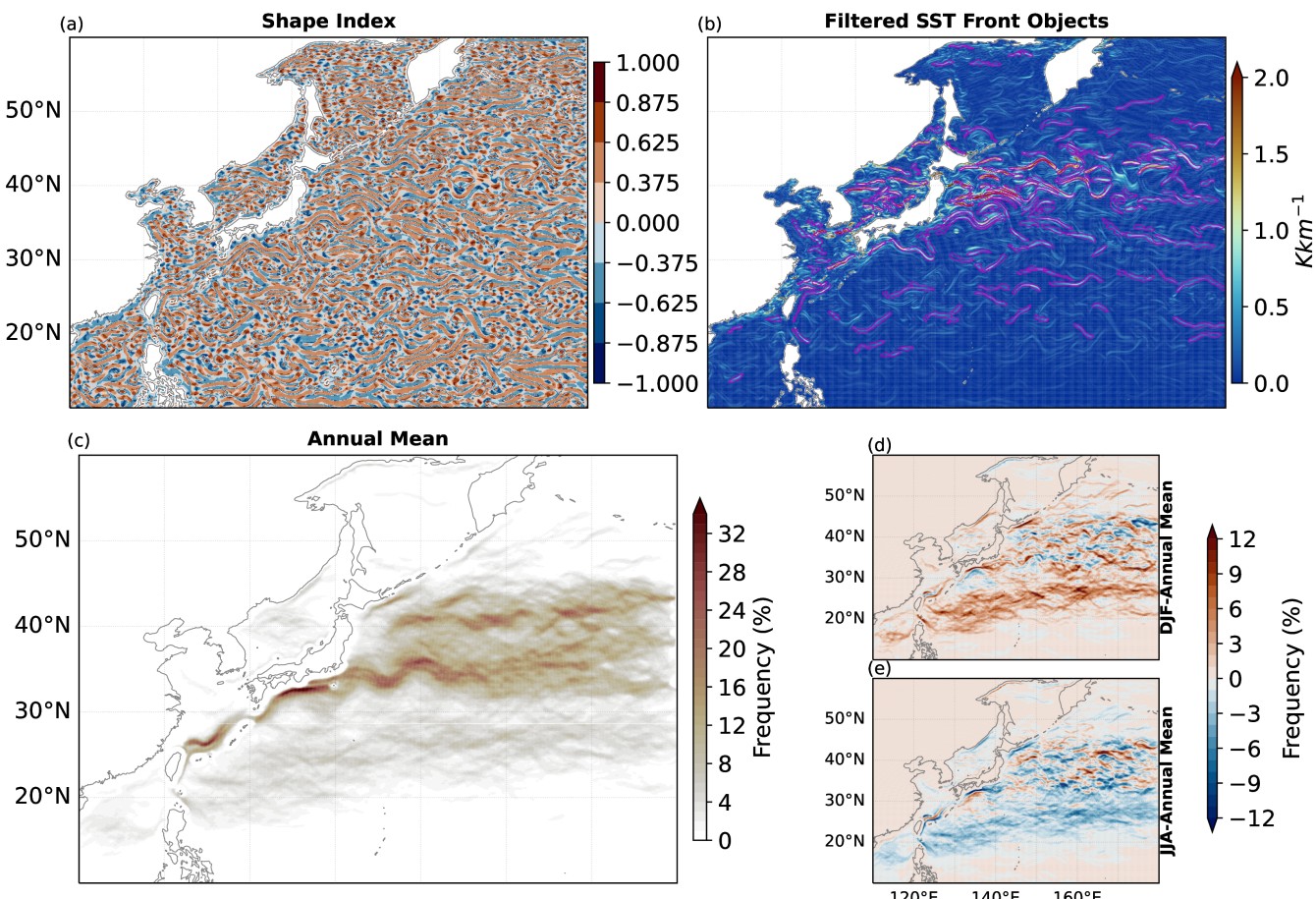

**Figure 6.** Major steps in the detection of sea surface temperature fronts (SSTFs). (a) Magnitude of SI as calculated from the smoothed primary field of the horizontal gradient of sea surface temperature ($\nabla$SST). Smoothing removes variability smaller than 15 km from $\nabla$SST. Filtered SSTF objects, in units of Kelvin per kilometer ($\mathrm{K\,km^{-1}}$), where background color represents unsmoothed values of $\nabla$SST. (c) Annual mean frequency of SSTF occurrence across a 10-year period in the present climate simulation. (d-e) Anomalous frontal frequencies relative to the annual mean for (d) JJA and (e) DJF.

## 4    Application to 3D Features

This section introduces the extension of SCAFET to detect features within three-dimensional (3D) primary fields. The process of scale-space selection involves applying Gaussian smoothing independently along each of the three dimensions. Notably, a 3D field yields three eigenvalues ($k_1 \geq k_2 \geq k_3$) instead of the usual two. In this context, the SI can be calculated in three different ways by combining these eigenvalues.

For the extraction of jet streams, the SI calculated using $k_1$ and $k_2$ (the two largest eigenvalues) is used as it provides a more conservative estimate for the jet-like structure (see Appendix subsection A3 and Supplementary Figure S7). The decision

to exclude the smallest eigenvalue, denoted as $k_3$, is based on empirical observations. Empirical evidence suggests that when dealing with regions exhibiting positive maxima (convex curvature), both $SI(k_1, k_2)$ and $SI(k_1, k_3)$ effectively capture the shape. Meanwhile, $SI(k_2, k_3)$ has a trivial application (refer to Figure A4). Conversely, for concave shapes, both $SI(k_1, k_3)$ and $SI(k_2, k_3)$ represent the shape, while the conditions for $SI(k_1, k_2)$ become redundant given that they are satisfied by $SI(k_1, k_3)$ and $SI(k_2, k_3)$.

## 4.1    Jet Streams

Jet streams, regardless of the underlying dynamics, are narrow, high-wind-speed regions in the upper atmosphere with faster wind speeds compared to their surroundings (Koch et al., 2006). These jet streams have a significant impact on aviation and strongly influence surface weather conditions. For example, a persistent jet stream in boreal summer can result in extreme heat and flooding events, while a meandering jet stream in winter leads to severe cold spells in the midlatitudes (Petoukhov et al., 2013; Coumou et al., 2014; Kretschmer et al., 2016). Additionally, the northward movement of jet streams due to greenhouse warming contributes to the poleward propagation of tropical cyclones (Studholme et al., 2021). Thus accurately detecting and characterizing jet streams is crucial for predicting and projecting both climatology and extreme weather systems.

Much like the detection of other weather phenomena discussed in this study, previous research typically employs a physical threshold to identify jet streams. Furthermore, with the exceptions of Limbach et al. (2012) and Kern et al. (2018), most studies identify jet streams as either one or two-dimensional features. However, it is important to emphasize that this section's focus is primarily on illustrating the method for detecting jet streams rather than validation of any analysis with published work. There is currently limited analysis available for comparing with a 3D perspective of jet streams, highlighting the need for such an approach. As a result, we present examples of jet stream detection in three selected time steps. A more comprehensive analysis and discussion regarding of the long-term characteristics of jet streams will be a topic for future research. For those interested, a video showcasing the results over an extended period can be found in the supplementary section.

The primary field used in the extraction of jet streams is the 6-hourly, three-dimensional wind speeds obtained from ERA5 reanalysis data set, with a spatial resolution of $1°$ with 37 vertical levels (Hersbach et al., 2020). The magnitude of wind speed is calculated as,

$$W = \sqrt{U^2 + V^2} \tag{8}$$

where U and V are the zonal and meridional wind velocities.

The detection process for jet streams begins similarly to the detection of 2D features. Gaussian smoothing is used to remove variability less than 3000 km in the horizontal dimensions. No smoothing is applied along the vertical dimension. Next, SI is calculated using the two largest eigenvalues, $k_1$ and $k_2$. The vertical dimension for the three-dimensional wind speed is given in pressure coordinates. To calculate the gradient as change in wind speeds per kilometer, a rudimentary conversion from pressure to height coordinates is used (refer to Wallace and Hobbs (1977, pg. 60-61), and https://unidata.github.io/MetPy/latest/api/generated/metpy.calc.pressure_to_height_std.html for further details).

| | No. | Property | Value | Unit |
|---|---|---|---|---|
| **Segmentation** | 1 | Smooth Scale | 6000 | km |
| | 2 | Selected Shape | (0.375,1.0] | - |
| **Filtering** | 1 | Minimum Length | 5000 | km |
| | 2 | Minimum Height | 5 | km |
| | 3 | Minimum Volume | $10^6$ | $km^3$ |
| | 4 | Minimum Max. Wind speed | 50 | $m\,s^{-1}$ |

**Table 4.** Same as in Table 1 but for parameters and values relevant for detecting jet streams.

Similar to the detection of ARs, regions characterized by the selected shapes of ridges, caps, and domes (SI > 0.375) are
isolated for filtering. Filtering is then applied to remove objects with a volume less than $10^6 km^3$, a horizontal length less than
5000 km, and a maximum wind speed within each object less than $50\,m\,s^{-1}$. In the current version of SCAFET, the tracking
algorithm is not applied to jet detection (see Figure 7). The detailed list of parameters used in the detection of jet streams is
given in Table 4.

## 5 Conclusions

In this study, we introduced a novel computational mathematical framework and an open-source Python package for extracting
and tracking features from large climate datasets, called Scalable Feature Extraction and Tracking (SCAFET). The purpose of
SCAFET is to tackle the challenges posed by the increasing volume and diversity of climate data by providing an alternative to
traditional physical threshold-based feature detection methods. It enables the comparison of features between observational and
model data with different mean states by attempting to remove the need for posterior data-specific assumptions. Furthermore,
SCAFET introduces a novel shape-based approach to feature extraction, which helps uncover discrepancies in climate projec-
tions due to differences in detection methods and aims to help the community in building scientific consensus. To demonstrate
SCAFET's capabilities and its potential in advancing these goals, we showcased its ability to detect various features, including
two-dimensional features such as atmospheric rivers (ARs), tropical and extratropical cyclones, sea surface temperature fronts,
as well as the detection of three-dimensional jet streams. Each application serves as an illustrative example from which users
can customize SCAFET for their specific research needs.

SCAFET offers several significant advantages, including a more comprehensive framework and parallel computing im-
plementation for efficiency. However, its most noteworthy contribution lies in offering a novel perspective on how we can

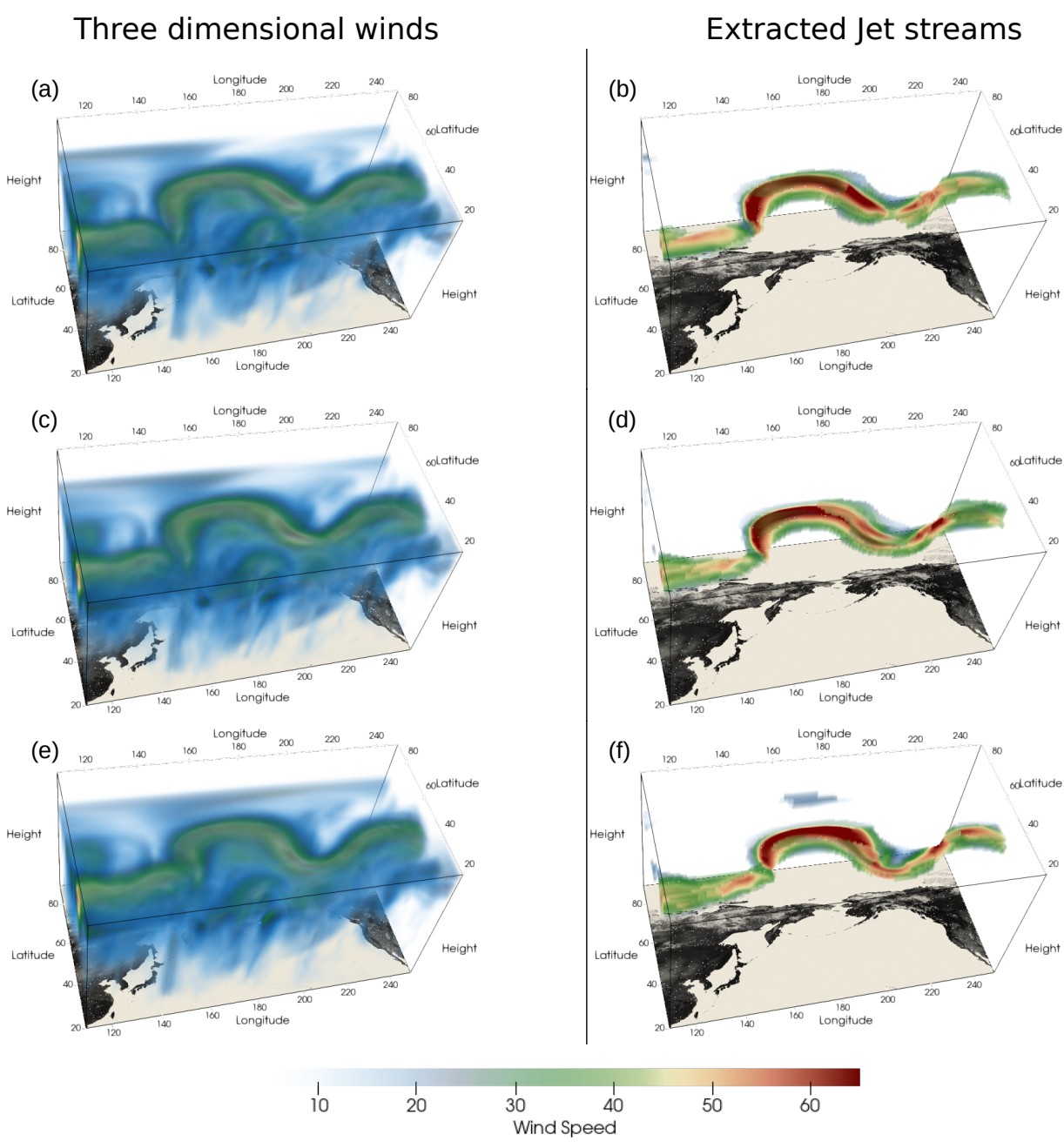

**Figure 7.** 3D jet streams extracted using SCAFET. Magnitude of 3D wind speed for (a) 2022-08-28 12:00, (c) 2022-08-28 18:00, and (e) 2022-08-29 00:00. Extracted 3D jet streams for corresponding periods are shown in (b), (d), and (f) respectively. The reader is encouraged to view the full video of these snapshots in the supplementary information.

relatively define various features within climate datasets that span extensive periods marked by significant changes in mean climate. Rather than relying on empirically-derived, data-specific physical thresholds for feature extraction, SCAFET identifies features using shape-based absolute thresholds and the locally estimated shape within the field. This methodology offers a unique viewpoint, enabling us to observe the continuous changes in feature properties while accounting for shifts in the mean climate state. This approach is particularly valuable as meso-synoptic scale studies are highly sensitive to thresholds in a dynamically changing mean climate state. Consequently, the conclusions drawn from such studies can vary significantly, as demonstrated in research examining the response of ARs to greenhouse warming (Zhao, 2020; O'Brien et al., 2022; Nellikkattil et al., 2023). Thus, algorithms like SCAFET which are not influenced by data-specific conditions of various climate models play a crucial role in advancing scientific understanding and facilitating climate model development.

In conclusion, delving deeper into the principles of differential geometry to elucidate the physical interpretation of the relationship between SI and local geometric shape has the potential to revolutionize our approach to feature extraction from large datasets. This avenue of research has the promise of significantly enhancing the algorithm's robustness and reliability. It's worth noting that, at present, SCAFET may not surpass the computational efficiency of other well-established feature extraction methods discussed above (see Supplementary Section 2.2). However, ongoing efforts to optimize and streamline the algorithm for improved computational efficiency continue. One notable strength of SCAFET's design is its independence from dataset-specific posterior information when identifying features. Moreover, the shape-based thresholds used for detecting specific features remain consistent across various grids, datasets, and climatologies. Between these strengths and the full parallelization of the feature detection method, there are exciting possibilities for further development. This may eventually enable the algorithm to be used in operational feature identification and early-warning systems for extreme weather events.

*Author contributions.* ABN wrote and developed the software package. ABN and DL prepared the manuscript draft with inputs from JYL. TAO and DL were involved in developing a mathematical framework for the algorithm. JEC provided input and guidance on the detection and tracking of tropical and extratropical cyclones. JYL, TAO, and JEC contributed equally to the manuscript revisions.

*Code and data availability.* The latest version of the Scalable Feature Extraction and Tracking (SCAFET) algorithm can be downloaded from https://github.com/nbarjun/SCAFET. The version of the codes used for feature extraction and creating relevant figures in this manuscript can be downloaded from https://doi.org/10.5281/zenodo.7767301. A sample dataset for the curvilinear SST data is also included in the repository. The directory also includes sample outputs for various features discussed in the manuscript. The ERA 5 reanalysis data with varying resolutions can be downloaded from https://cds.climate.copernicus.eu/cdsapp. Single-level variables like 10m wind are obtained from https://cds.climate.copernicus.eu/cdsapp#!/dataset/reanalysis-era5-single-levels?tab=form, while three-dimensional variables can be extracted from https://cds.climate.copernicus.eu/cdsapp#!/dataset/reanalysis-era5-pressure-levels?tab=form. To see the exact codes used for downloading ERA5 data, readers could refer to the *ERA5Data* folder in the Zenodo repository. For any further details on code and data, feel free to contact the corresponding author.

*Competing interests.* TAO is a member of the editorial board of the journal Geoscientific Model Development. The peer-review process was
guided by an independent editor, and the authors have no other competing interests to declare.

*Acknowledgements.* The research was supported by the Institute for Basic Science (IBS), Republic of Korea, under IBS-R028-D1. Lemmon's
contributions were in part supported by their status as a Science and Technology Policy Fellow with the American Association for the
Advancement of Science. O'Brien's contributions were supported by the Director, Office of Science, Office of Biological and Environmental
Research of the U.S. Department of Energy under Contract No. DE-AC02- 05CH11231 and by the Environmental Resilience Institute,
funded by Indiana University's Prepared for Environmental Change Grand Challenge initiative. The authors would like to thank Prof. Axel
Timmermann, Dr. Karl Stein, Dr. Ryohei Yamaguchi, and Dr. Pavan Harika Ravi for their comments on shape-based feature extraction. We
would also like to thank the ARTMIP community for their feedback on SCAFET. Part of the analysis of the detected features was conducted
on the IBS/ICCP supercomputer "Aleph" 1.43 petaflops high-performance Cray XC50-LC Skylake computing system with 18,720 processor
cores, 9.59 PB storage, and 43 PB tape archive space. We also acknowledge the support of KREONET for the fast and reliable data transfers.
Lastly, we extend special thanks to early users of the algorithm.

## Appendix A: Shape Based Feature Extraction on Simple Datasets

This section demonstrates how shape-based feature extraction can be performed on scalar fields represented by simple, ideal-
ized mathematical functions. It is intended to provide readers with more insights into the basic principles behind shape-based
feature extraction and how it differs from other conventional methods. We have also tried to showcase some properties of
shape-based feature extraction methods like its insensitivity to linear mean state trends.

### A1    Application to 1D datasets

In this section, we draw an analogy between the use of SCAFET on a two-dimensional (2D) dataset and shape-based feature
extraction from a one-dimensional (1D) dataset. Our intention is not to promote the use of shape-based extraction of features
from 1D datasets but rather to provide readers with a fundamental understanding of this approach, along with its strengths and
limitations.

For any differentiable curve $C$, the curvature is measured as the instantaneous rate of change of direction along the curve.
Simply put, the curvature is measured as the rate of change of the unit tangent to the curve at any given point. An osculating
circle can be used to intuitively represent the curvature of a surface or a curve (see Figure A1). At any point P, the curvature,
$k$ is the reciprocal of the radius ($R$) of the circle. The sign of $k$ determines if the curve has a concave or a convex curvature.
More information and mathematical proof for these concepts can be found in any standard differential geometry textbook.

Following the derivation of Shape Index (SI) for 2D datasets, we calculate the local shape of a function $f$ using the shape
parameter, defined as

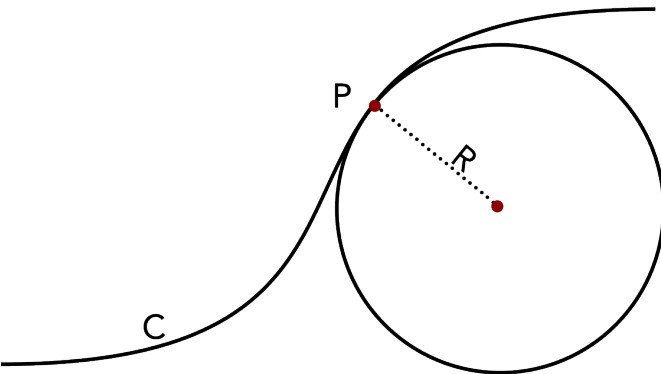

**Figure A1.** Schematic representation of curvature measurement of a curve C at point P. At P, the curvature is the reciprocal of the radius R of the osculating circle. In differential geometry, an osculating circle is defined as the circle passing through the point P and a pair of additional points infinitesimally close to P.

$$K = \frac{2}{\pi} \tan^{-1}(f'') \qquad \text{(A1)}$$

Values of $K$ closer to 1 are identified as regions of local minima while $K$ closer to -1 are regions of local maxima (black curve in Figure A2). Depending on the magnitude of the function, one could adjust the value of $K$ to obtain regions of local maxima (red caps in Figure A2) and local minima (green caps in Figure A2). The curvature of the function is insensitive to linear trends and mean state changes. This is evident as the application of identical shape thresholds identifies the same regions of the curves as local maxima and minima, whether on the base curve (blue curve in Figure A2) or on the same curve with an added linear trend (orange curve in Figure A2). The values of $K$ for both curves are represented by the black line in Figure A2. Thus, the shape parameter can be used to identify the local minima and maxima from a 1D dataset despite background state changes.

## A2   Application of SCAFET to simple Geostrophic Motion

In this section, we apply SCAFET to a basic geostrophic rotational motion. The goal of this discussion is to illustrate how the shape-based extraction of 2D features differs from conventional methods. The calculation of SI involves the computation of the two eigenvalues, $k_1$ and $k_2$ of the Hessian matrix for any gridded dataset. As discussed in the previous section, the curvature measurement provided by $k_1$ and $k_2$ can be visualized as the reciprocal of the radius of two osculating circles that intersect orthogonally at a point on the surface. Large negative eigenvalues signify surfaces with strong convex curvature, while positive values correspond to troughs or cups.

To demonstrate the characteristics and advantages of feature detection based on SI, let's consider a simple rotational wind field (see Figure A3(a) vectors) given by,

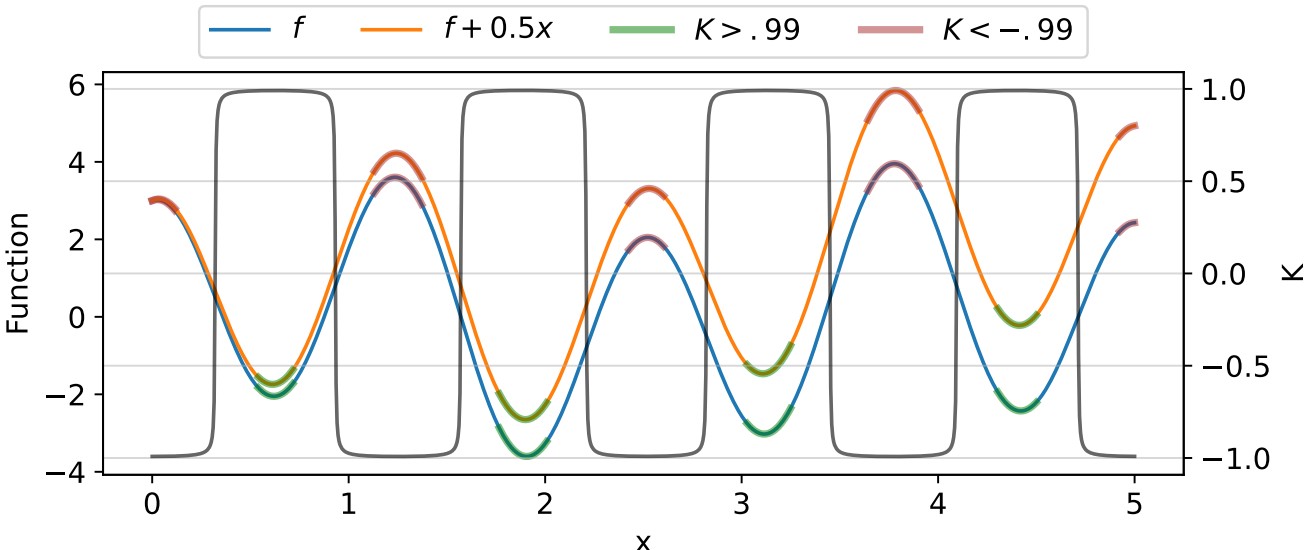

**Figure A2.** Comparison of shape extraction between a simple one-dimensional curve, given by $f = \sin 2x + 3\cos 5x$ (blue curve) and $f + 0.5x = \sin 2x + 3\cos 5x + 0.5x$ (orange; the blue curve with a linear trend). Left Y-axis shows magnitude of both functions, right Y-axis indicates values of the shape parameter ($K$). Note that value of $K$ is the same for both functions. The green and red highlighting on the curves shows regions where $K > 0.99$ and $K < -.99$, corresponding to regions of local maxima and minima, respectively.

$$u_g = -\Omega y \tag{A2}$$

$$v_g = \Omega x \tag{A3}$$

Where $\Omega$ is a constant ($\Omega = 10^5 rads^{-1}$) and x,y represents the grid. The geopotential height ($h$) of the field (see Figure A3(a) shading) is used as our primary field in calculating SI, computed as,

$$h = \frac{\Omega f}{2g}(x^2 + y^2) \tag{A4}$$

where $f$ and $g$ are the Coriolis parameter and the acceleration due to gravity, respectively. SI is calculated from the eigenvalues of the Hessian of $h$ using the formula,

$$SI(k_1, k_2) = \frac{2}{\pi} \tan^{-1}\left[\frac{k_2 + k_1}{k_2 - k_1}\right] \tag{A5}$$

Where the eigenvalues $k_1$ and $k_2$ are given by,

$$455 \quad k_{12} = \frac{f\zeta_g}{2g} \pm \sqrt{\left(\frac{f}{2g}\right)^2 - \left(\frac{f}{g}\right)^2 \frac{\partial v_g}{\partial x}\frac{\partial u_g}{\partial y} + \frac{\partial u_g}{\partial x}\frac{\partial v_g}{\partial y}} \tag{A6}$$

Where $\zeta_g$ is the geostrophic vorticity. Which gives SI as,

$$SI(k_1, k_2) = \frac{2}{\pi}\tan^{-1}\left[\frac{\zeta_g}{-2\sqrt{\left(\frac{\zeta_g}{2}\right)^2 - \frac{\partial v_g}{\partial x}\frac{\partial u_g}{\partial y} + \frac{\partial u_g}{\partial x}\frac{\partial v_g}{\partial y}}}\right] \tag{A7}$$

A detailed derivation of the above equation can be found in Appendix B Plugging in the values for the rotational motion, we get

$$460 \quad \zeta_g = \nabla^2 h = \Omega f/g \tag{A8}$$

$$\frac{\partial u_g}{\partial x} = \frac{\partial v_g}{\partial y} = 0 \tag{A9}$$

$$\frac{\partial v_g}{\partial x}\frac{\partial u_g}{\partial y} = \Omega^2 \tag{A10}$$

Therefore,

$$SI = \frac{2}{\pi}\tan^{-1}\left[\frac{\Omega^2}{-\sqrt{\Omega^2 - \Omega^2}}\right] = -1 \tag{A11}$$

Thus, SCAFET classifies the whole domain with anticlockwise rotational motion as a trough with SI≈-1 regardless of the absolute value of the field or $\Omega$. In contrast, traditional methods that rely on thresholding the geopotential height would identify regions based on the chosen threshold of $h$, which would need to be adjusted depending on the mean (time) and background (space) state. Another common approach is to establish a threshold on the smallest eigenvalue, aiming to identify extreme features based on the curvature strength rather than the field's actual value. TempestExtremes (Ullrich and Zarzycki,

2017), a feature extraction framework discussed in the main text, follows this method to detect Atmospheric Rivers from gridded datasets. In the current example, this approach would correspond to setting a threshold on $f\Omega/g$. In other words, TempestExtremes would only identify a trough if the value of $\Omega$ exceeds the pre-determined threshold. SCAFET, on the other hand, identifies the trough region as a trough regardless of the specific value of the field or $\Omega$. This illustrates how feature extraction using SI and other published methods can yield different results depending on the input data, as they focus on

distinct properties of the field.

## A3   Application of SCAFET to 3D Fields

This section aims to demonstrate the detection of a cylindrical volume within a three-dimensional scalar field. To illustrate the effectiveness of the SI in identifying 3D structures embedded within scalar fields, we offer a straightforward example of how SI

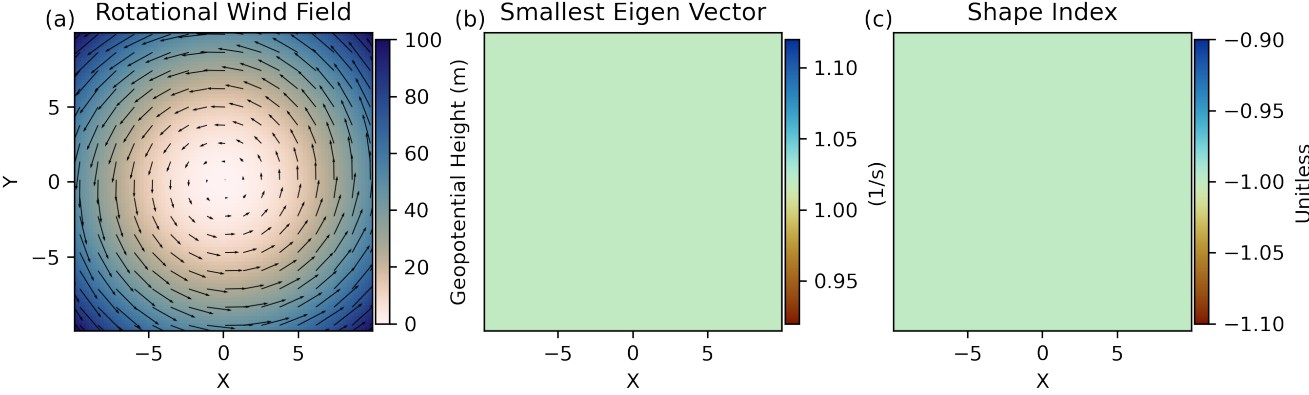

**Figure A3.** Comparison between two feature extraction techniques on an idealized example of rotational wind field. (a) The geopotential height ($h$) (shading) of the rotational wind field (arrows). h is defined as $\Omega f(x^2 + y^2)/2g$, where $f = 10^{-4}s^{-1}$, $g = 9.805\,m\,s^{-1}$ and $\Omega = 10^5 rad\,s^{-1}$. (b) Magnitude of the smallest eigenvalue, derived from the equation as $f\Omega/g = 1.0199$, illustrating a uniform field as expected. (c) Value of the SI where $SI = -1$ throughout the domain, as expected.

can be used to isolate a cylinder embedded in a scalar field defined by $f = \sin(3X) + \cos(4Y)\cos(Z)$. It is worth mentioning that this specific problem bears significant similarities to the task of identifying 3D jet cores.

As explained in section 4, a three-dimensional field provides us with three eigenvalues satisfying the condition $k_1 \geq k_2 \geq k_3$. The SI can be computed using $SI(k_1, k_2)$, $SI(k_1, k_3)$, or $SI(k_2, k_3)$. Setting a threshold of SI > 0.375 effectively isolates the cylinder when using either $SI(k_1, k_2)$ or $SI(k_1, k_3)$ (see Figure A4(b-d)). Between these two options, $SI(k_1, k_2)$, which utilizes the two largest eigenvalues, imposes a more conservative criterion for identifying the embedded cylinder. The percentage of data identified as the cylinder is provided in the title of each plot in Figure A4. Notably, employing $SI(k_2, k_3)$ is not suitable as it fails to isolate the desired cylinder shape effectively. The choice of using $SI(k_1, k_2)$ is specifically tailored for extracting convex shapes or local maxima. Interestingly, to identify concave shapes or local minima, one should utilize the SI derived from the two smallest eigenvalues, namely, $SI(k_2, k_3)$.

While the simple example presented here may not provide a comprehensive illustration of 3D feature detection, we hope that it encourages further fundamental research into 3D feature extraction to expand the capabilities of analysis and increase precision.

**Figure A4.** Various approaches for extraction of a 3D cylinder from a scalar field. (a)Simple scalar field represented by $\sin(3X) + \cos(4Y) * \cos(Z)$ is shown. (b-d) The extracted cylinders by applying the conditions (b) $SI(k_1, k_2) > 0.375$, while in (c) $SI(k_1, k_3) > 0.375$ and (d) $SI(k_2, k_3) > 0.375$ are shown. The values enclosed in parentheses within the figure titles indicate the percentage of data that satisfies the respective conditions applied in each case.

## Appendix B: Derivation of Shape Index for Geostrophic Motion

The complete derivation of the SI for geostrophic wind fields is shown in this section. The result from the derivation is used in Appendix A.

Let $h$ be the geopotential height at a certain level. The Hessian of h is given by

$$\mathcal{H}(h) = \begin{pmatrix} \frac{\partial^2 h}{\partial x^2} & \frac{\partial^2 h}{\partial x \partial y} \\ \\ \frac{\partial^2 h}{\partial y \partial x} & \frac{\partial^2 h}{\partial y^2} \end{pmatrix} \tag{B1}$$

The eigenvalues of the symmetric matrix $\mathcal{H}$ is calculated by solving the quadratic equation.

$$\left( \frac{\partial^2 h}{\partial x^2} - \lambda \right) \left( \frac{\partial^2 h}{\partial y^2} - \lambda \right) - \left( \frac{\partial^2 h}{\partial x \partial y} \right)^2 = 0 \tag{B2}$$

which can be expanded as;

$$\lambda^2 - \lambda \left( \frac{\partial^2 h}{\partial x^2} + \frac{\partial^2 h}{\partial y^2} \right) + \frac{\partial^2 h}{\partial x^2} \cdot \frac{\partial^2 h}{\partial y^2} - \left( \frac{\partial^2 h}{\partial x \partial y} \right)^2 = 0 \tag{B3}$$

$$\lambda^2 - \lambda \nabla^2 h + \frac{\partial^2 h}{\partial x^2} \cdot \frac{\partial^2 h}{\partial y^2} - \left( \frac{\partial^2 h}{\partial x \partial y} \right)^2 = 0 \tag{B4}$$

NOTE: The geostrophic vorticity ($\zeta_g$) is defined as

$$\zeta_g = \frac{g}{f} \nabla^2 h \tag{B5}$$

The geostrophic velocities are defined as

$$u_g = -\frac{g}{f}\frac{\partial h}{\partial y} = -\frac{\partial \psi}{\partial y} \tag{B6}$$

$$v_g = \frac{g}{f}\frac{\partial h}{\partial x} = \frac{\partial \psi}{\partial x} \tag{B7}$$

Where $\psi$ is the geostrophic stream function. This implies.

$$\frac{\partial^2 h}{\partial x^2} = \frac{f}{g}\frac{\partial^2 \psi}{\partial x^2} = \frac{f}{g}\frac{\partial v_g}{\partial x} \tag{B8}$$

$$\frac{\partial^2 h}{\partial y^2} = \frac{f}{g}\frac{\partial^2 \psi}{\partial y^2} = -\frac{f}{g}\frac{\partial u_g}{\partial y} \tag{B9}$$

Adding the abovementioned relationships to equation (3)

$$\lambda^2 - \frac{\lambda f}{g}\zeta_g - \frac{f^2}{g^2}\frac{\partial v_g}{\partial x}\frac{\partial u_g}{\partial y} + \frac{f^2}{g^2}\frac{\partial u_g}{\partial x}\frac{\partial v_g}{\partial y} \tag{B10}$$

Solving for $\lambda$ we get

$$\lambda_{12} = \frac{f\zeta_g}{2g} \pm \sqrt{\left(\frac{f}{2g}\right)^2 - \left(\frac{f}{g}\right)^2 \frac{\partial v_g}{\partial x}\frac{\partial u_g}{\partial y} + \frac{\partial u_g}{\partial x}\frac{\partial v_g}{\partial y}} \tag{B11}$$

$$\lambda_{12} = \frac{f}{g}\left[\frac{\zeta_g}{2} \pm \sqrt{\left(\frac{\zeta_g}{2}\right)^2 - \frac{\partial v_g}{\partial x}\frac{\partial u_g}{\partial y} + \frac{\partial u_g}{\partial x}\frac{\partial v_g}{\partial y}}\right] \tag{B12}$$

Thus the shape index for $h$

$$SI = \frac{2}{\pi}\tan\left[\frac{\zeta_g}{-2\sqrt{\left(\frac{\zeta_g}{2}\right)^2 - \frac{\partial v_g}{\partial x}\frac{\partial u_g}{\partial y} + \frac{\partial u_g}{\partial x}\frac{\partial v_g}{\partial y}}}\right] \tag{B13}$$

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
