# Peer review of "Scalable Feature Extraction and Tracking (SCAFET): A general framework for feature extraction from large climate datasets"

_EGUsphere, 2023_

## Referee Comment (RC2)

**Scalable feature extraction and tracking (SCAFET): A general framework for feature extraction from large climate datasets**

https://doi.org/10.5194/egusphere-2023-592
EGU Sphere
July 10, 2023

**1 Summary**

A feature extraction algorithm is presented that combines elements of shape recognition with existing feature extraction methods based on more traditional atmospheric and oceanic variables. The utility of the method is demonstrated on 2D problems that include atmospheric rivers, tropical cyclones, and oceanic SST fronts. A 3D demonstration (jet stream analysis) is also presented.

The shape recognition procedure is based on an analysis of a smoothed data surface field using properties of the second order multidimensional Taylor series expansion, which in includes the local Hessian matrix of the surface. Such a technique has the potential to introduce an objectively defined qualitative "shape" into a data-based algorithm for feature detection, which is a significant advance. The variety of the chosen demonstration applications shows the high potential of the method for the broad field of climate modeling.

Unfortunately, the paper includes seemingly contradictory statements, statements that are not justified by the presented work, and omits some qualifications related to the method's practical use. For example, the introduction lists many challenging pieces of the climate data analysis problem and suggests that the presented method is a solution to them. These challenges include such things as objective identification criteria that are independent of specific model configurations, a reduced need for preprocessing data, and the challenge of dealing with enormous data output from high resolution simulations. When the results are presented in subsequent sections, however, the reader discovers that the filtering kernel must be tuned to a particular grid and that several of the standard preprocessing steps are still necessary. The computational performance of the method, relevant for the large data discussion, is not demonstrated. I therefore recommend major revisions.

The presented work has significant potential to become a new standard for climate data analysis, and I encourage the authors to continue its development.

**2 General comments**

1. Figure 1: The smoothing scale $\sigma$ is defined as a function of the grid and the length scale of the feature of interest; however, this function is not defined in the paper. In the sentence beginning on line 131, the statement "This is implemented by calculating the value of $\sigma$ along each circle of latitude," is particularly uninformative.

2. The introduction spends a lot of time and effort making a case for objective detection methods that do not rely on a human that will work well despite the challenge of "inter- and intra-model discrepancies" (line 38). Having to adapt the smoothing kernel to the grid spacing of specific resolution configurations seems to undercut those primary goals.

3. Koenderink and van Doorn (1992) advocate the use of pair of measures for shape recognition, "curvedness" and "shape index." The present work seems to discard "curvedness" but does not mention why. Furthermore, it does not discuss significant considerations related to the use of these measures for meteorological applications.

   The Hessian of a surface, $z(x, y)$, is a second-order term in the surface's local Taylor series expansion; its use here is only applicable if the first-order term (the gradient of $z(x, y)$) is zero. Indeed, even the

cited reference that the authors rely upon, Koenderink and van Doorn (1992), contains the strong caution that the interpretation of shape from the Hessian matrix (emphasis theirs) "is *only* valid in representations where the magnitude of the gradient of $z$ vanishes." This would seem to suggest that a preprocessing step to find critical points (where the gradient is zero) is necessary, but I did not see such a step mentioned.

4. The discussion of the method's extension to 3D needs more detail. Given the horizontal-vertical splitting common to climate data sets, this is a nontrivial task. All of the shape index mathematics are formulated for 2D problems (with only 2 eigenvalues); for the 3D demonstration, the choice to use $k_1$ and $k_3$ and exclude $k_2$ seems significant. What is its physical interpretation? Perhaps this is justifiably outside the scope of the paper, but I wonder how would the definition of shape index change to account for all three eigenvalues? What are the analogous "shapes?"

5. The notation in equation (2) is confusing. It is customary for the numerator to contain the maximum order of each derivative; here, in all four terms the superscript 2 is missing. In the mixed terms, the symbol $\partial$ is missing between the $x$ and $y$. The same comment applies to the indices.

6. I prefer to see the functional dependence of newly introduced methods defined explicitly; for example in Equation (3), $SI(k_1, k_2)$ is more informative than simply $SI$, and $\phi_s(x, y)$ should match $\phi_p(x, y, \dots)$ in equation (1).

7. What to the dots ($\dots$) represent in equation (1), and in Figure 1's definition of $\phi_s$ in the exponent?

8. The EGU audience is interdisciplinary, and some may not be as familiar with the properties of the Hessian matrix and its eigenvectors. Adding their illustration to the dictionary of shapes in Figure 2 would be very helpful.

9. It's not clear how distinct and/or subjective the boundaries between different SI regions are; for example, how different is a "Rut" with SI = -0.374 from a "Saddle Point" with SI=-0.375? What about other boundaries, e.g., Ridges and Saddles, Ridges and Caps/Domes? How does this affect the various features that are sought — how easily could a "cap" be misclassified as a "ridge," and how significant might that be to the results of a study?

10. A validation study comparing SCAFET to existing methods (such as TECA, which the authors mention) would be helpful — given the same criteria, do they detect the same tropical cyclones? If there are differences, what are the characteristics of the storms that appear in one but not the other. Similarly, given the emphasis of the work on large data sets produced by high resolution models, how does the computational performance of the proposed method compare with previous methods? Does the method achieve faster processing times? Both variety of studies were performed in [1], which should also be cited here.

11. The paragraph beginning on Line 35 suggests that defining thresholds for particular features is challenging and can vary between and even within individual models. The implied suggestion is that the presented method, SCAFET, would solve this problem; however, the remainder of the work relies on the same expert analysis (for example, Table 1) that the paper claims to avoid elsewhere. Similarly, Table 2 presents a set of well-defined criteria for tropical cyclones that contradicts the Line 35 paragraph.

12. Consider re-drawing Figure 7 with a white background for hard-copy readers.

**3   Specific comments**

1. Figure 1: "Hessian" should be capitalized.

2. There are numerous grammatical and typesetting errors. An incomplete list includes:

   (a) The inverse tangent function in equation (3) should not be italicized; so should sin and cos in the Figure 2 caption.

   (b) Period missing after equation (3).

   (c) The symbols $n$ and $n + 1$ in Section 2.3 should be italicized.

   (d) "circ" in Lines 202 and 236 should be °.

   (e) km in lines 203, 204 should not be italicized

   (f) Please use either $\tan^{-1}$ or arctan, not both.

**References**

[1] P. A. Bosler, E. L. Roesler, M. A. Taylor, and M. R. Mundt, 2016, Stride Search: A general algorithm for storm detection in high-resolution climate data, *Geoscientific Model Development* 9:1383–1398.

---

## Author Comment (AC1)

Arjun Babu Nellikkattil,

Ph.D. Candidate, Pusan National University,

Institute of Basic Science, Center for Climate Physics,

Email: arjunbabun@pusan.ac.kr

August 20, 2023

**Re: Minor revisions of manuscript EGUSPHERE-2023-592**

Dear Reviewer,

Thank you sincerely for your valuable suggestions and kind words of appreciation. My co-authors and I highly appreciate the importance and insights provided in your comments. We have made genuine efforts to address each of your suggestions in the revised manuscript.

Below, you will find a point-by-point response outlining our approach to your suggestions and comments. Additionally, we have introduced a new section within the response letter, emphasizing the major revisions that have been executed in response to all reviewers' recommendations. Once the editor approves these responses, we would submit a revised version of the manuscript as per the *Geoscientific Model Development* guidelines.

Once again, we extend our gratitude for your thorough review and constructive feedback. Please do not hesitate to convey any further concerns or suggestions you may have; your input is crucial in enhancing the quality and rigor of our work.

Sincerely,

Arjun Babu Nellikkattil

**Contents**

**1 Recommendation**

SCAFET is a significant advance over the traditional absolute thresholding methods currently used by climate practitioners. With some minor revisions, see below, I recommend the manuscript for publication.

**2 Revision Highlights**

1. An additional appendix section has been included to provide a discussion and demonstration of the detection of three-dimensional structures within embedded 3D scalar fields.

2. An additional supplementary section has been added, presenting the sensitivity analysis of various parameters in the detection and characteristics of Atmospheric Rivers. The primary objective of this section is to illustrate the stability and reliability of the diverse parameters employed in SCAFET.

3. An extra supplementary section has been incorporated, offering a comparison of the identified characteristics of Atmospheric Rivers. Additionally, the computational aspects of SCAFET are contrasted with those of other detection algorithms.

4. The manuscript has been further refined to enhance grammar and comprehension.

**3 Point-by-point Response**

1. My main comment or question is related to how sensitive feature identification is to SCAFET parameters. You have shown that it is possible to identify weather features with SCAFET, which is great, but there is no discussion on how sensitive the results are. For example, how sensitive is the detection of ARs in Figure 4 to the parameters used in Table 1? On the one hand, it is intuitive to identify ARs as long, narrow shapes with (relatively) high IVT and precipitation. But concrete numbers must be used to implement that intuition. If you slightly change the SI threshold for Ridges, or the minimum length, or angle coherence, etc. does this totally change the kind of objects identified so that they no longer resemble ARs (I wouldn't think so, but perhaps), or does it slightly change the details of ARs detected? If it is the latter case, how did you decide on the exact values used in Table 1 for the best identification of ARs? I see there is one sentence, "The quantitative values for the properties are obtained from a consensus of previous studies referenced within each section." but I think this requires more elaboration.

We appreciate your comment and concur that a more in-depth discussion regarding the sensitivity of the employed parameters for feature extraction is indeed essential. Given the length constraints of the manuscript, we have integrated this discussion as a new supplementary section.

This section focuses on exploring the sensitivity of various parameters in detecting Atmospheric Rivers (ARs) using SCAFET. Specifically, we examine how modifications in the detection parameters influence key AR characteristics such as mean frequency, size, and count of ARs. Furthermore, the section incorporates a brief discussion explaining the observed sensitivity.

To facilitate readers' access to this supplementary section, an additional sentence has been introduced in Section 3.1, directing them to this content.

2. My second question is, what are we supposed to take away from Section 4.1 on Jet Streams? It shows some proof-of-concept that the method can be applied, in principle, to 3D data. To my eye, I don't see a clear jet stream identified by SCAFET in (b), (d), and (f) of Figure 7. So while the method can be applied to 3D data, it is not clear that it is successful in identifying features in 3D data.

In response to the feedback from Reviewer 2, the authors have made improvements to address this issue. We have added a new appendix section to showcase the identification of 3D cylinders embedded in simple scalar fields. This approach is analogous to the simpler examples demonstrated for 1D and 2D feature detection in Appendix A. The principal intent of this section is to underscore SCAFET's adeptness in accurately recognizing 3D structures.

The authors acknowledge the potential challenge in interpreting the presentation of 3D feature identification. They posit that this complexity stems from the intricacies inherent in effectively plotting and visualizing three-dimensional fields. To rectify this, we have made modifications to Figure 7. Specifically, they have chosen to focus on just one of the upper level jets, which could potentially make it easier to discern and evaluate the effectiveness of the SCAFET in identifying jet streams.

3. Third, while I think SCAFET is indeed a significant advance, I believe there are some statements made in the paper which are not justified, or I have misunderstood what you are trying to say.

   (a) Around line 40 there is discussion of dataset pre-processing, such as computing IVT fields for AR detection, and how this becomes infeasible for high resolutions and large ensembles. At first I read this as implying pre-processing as a downside of traditional methods, but something that SCAFET would bypass. However, SCAFET itself uses these pre-processed fields in the identification of ARs and cyclones.

   The reviewer is correct in pointing out the potential issues with making such sweeping statements. We intended to highlight the process of deriving thresholds from the input dataset during the segmentation procedure. Consequently, the entire dataset, especially for the considered period, has to undergo preprocessing to calculate thresholds. In contrast, SCAFET operates on a per-timestep basis, and although standard processes like smoothing are involved, there is no need to preprocess the **entire dataset**. The advantage of this approach is that we avoid the need to recalculate thresholds depending on the time-period or dataset under consideration. In simple terms, SCAFET can identify features using just a single timestep, while many conventional methods would necessitate threshold derivation following the preprocessing of extensive representative datasets.

   The sentence has been revised to emphasize this distinction more clearly.

(b) Starting at the end of line 327, there is the sentence "Due to its design, SCAFET does not require a priori climate information to identify features." I am not sure what is meant by this sentence. In the work presented, the shape-based component is only one piece of the full pipeline to identify weather features. Most obvious, the shapes are extracted from pre-processed fields IVT and RV, which are created from "climate information". Even knowing what generic shapes are appropriate for particular weather features I see as climate information.

Yet another sweeping statement. Thank you for pointing it out. The sentence has been modified now.

4. Finally, the writing and grammar etc. of the paper need to be cleaned up. Below are some instances I found during my reading:

(a) line 15: "… and value (5Vs) of climate data (REFs) of climate data."

(b) starting in line 162: "In the current study, a simple radius is defined and the closest object within the given radius to each object at time n is clustered and identified from time n+1 as the same object in motion." I get the general idea of what you are saying here, but I found this sentence hard to parse.

(c) line 186-187: "…derive this threshold from dataset directly, …"

(d) line 204-205: "… each object is used as to filter …"

(e) line 274: "…, SSTFs are not tracked as ocean fronts are stationary rather than…"

(f) in the middle of the Figure 6 caption, "In the next step, ridges, caps, and domes are extracted from (b) and weak and small…", do you mean for this to be (a) instead of (b)?

(g) line 293-294: "Since the scope of this section is limited to the validation of the detection method, we have only shown jet detection in three selected time steps." I'm not fully sure what you are trying to say here. Do you mean that filtering and tracking steps have not been performed here?

(h) Figure 7 caption: "The 3D jet streams extracted for the corresponding time period is show in …"

(i) line 362-363: "change of direction of a along the curve."

(j) In Figure A2, please adjust the legend so it can be read more clearly

All the suggestions have been implemented. The manuscript has undergone further refinement to eliminate such grammatical errors. We appreciate your thorough review and attention to detail.

---

## Author Comment (AC2)

Arjun Babu Nellikkattil,

Ph.D. Candidate, Pusan National University,

Institute of Basic Science, Center for Climate Physics,

Email: arjunbabun@pusan.ac.kr

August 21, 2023

**Re: Minor revisions of manuscript EGUSPHERE-2023-592**

Dear Reviewer,

We genuinely appreciate your valuable suggestions and kind words. Your comments were highly insightful and rigorous, and we greatly appreciate the careful consideration you have given to our manuscript.

We have diligently worked to address each of your comments, and as a result, the manuscript has undergone significant quality enhancement. Below, we present a point-by-point response to your suggestions, outlining the specific changes we made in response to your feedback. Furthermore, we have included a new section in the response letter, highlighting the major revisions carried out in accordance with the all reviewer's recommendations. Upon securing the editor's approval for these responses, our intention is to submit the revised manuscript in accordance with the guidelines outlined by *Geoscientific Model Development*.

Once again, we wish to express our gratitude for your thorough review and constructive feedback. Should you have any further concerns or suggestions, please do not hesitate to share them with us; your continued engagement will undoubtedly contribute to the overall excellence and rigor of our research.

Thank you once more for your time and dedication to this peer review process.

Sincerely,
Arjun Babu Nellikkattil

**Contents**

**1 Recommendation**

The presented work has significant potential to become a new standard for climate data analysis, and I encourage the authors to continue its development.

**2 Revision Highlights**

1. An additional appendix section has been included to provide a discussion and demonstration of the detection of three-dimensional structures within embedded 3D scalar fields.

2. An additional supplementary section has been added, presenting the sensitivity analysis of various parameters in the detection and characteristics of Atmospheric Rivers. The primary objective of this section is to illustrate the stability and reliability of the diverse parameters employed in SCAFET.

3. An extra supplementary section has been incorporated, offering a comparison of the identified characteristics of Atmospheric Rivers. Additionally, the computational aspects of SCAFET are contrasted with those of other detection algorithms.

4. The manuscript has been further refined to enhance grammar and comprehension.

**3 Point-by-point Response to General Comments**

1. Figure 1: The smoothing scale $\sigma$ is defined as a function of the grid and the length scale of the feature of interest; however, this function is not defined in the paper. In the sentence beginning on line 131, the statement "This is implemented by calculating the value of $\sigma$ along each circle of latitude," is particularly uninformative.

   To perform Gaussian smoothing that is aware of the grid structure in terms of latitude and longitude, we begin by computing the appropriate $\sigma$ value for each latitude band. The scale for smoothing is provided in kilometers as input. This input is then converted into the corresponding $\sigma$ value using the grid information, which is initially provided as input. For each latitude line, the $\sigma$ value is recalculated and applied to achieve smoothing. Subsequently, a constant $\sigma$ value is used for smoothing along the lines of longitude. The smoothing process employs the smoothing function from MetPy, which can be found at this URL (https://unidata.github.io/MetPy/latest/api/generated/metpy.calc.smooth_gaussian.html).

2. The introduction spends a lot of time and effort making a case for objective detection methods that do not rely on a human that will work well despite the challenge of "inter- and intra-model discrepancies" (line 38). Having to adapt the smoothing kernel to the grid spacing of specific resolution configurations seems to undercut those primary goals.

   Indeed, if the smoothing parameter had to be manually tweaked for different grids, it would undermine the claim. However, the smoothing parameter is provided by the users in kilometers, which is independent of the grid. The code then internally constructs the smoothing parameter

($\sigma$), taking into account the grid information. This ensures that the method remains consistent and adaptable to different grids without the need for grid-specific adjustments.

3. Koenderink and van Doorn (1992) advocate the use of pair of measures for shape recognition, "curvedness" and "shape index." The present work seems to discard "curvedness" but does not mention why. Furthermore, it does not discuss significant considerations related to the use of these measures for meteorological applications.

   The fundamental principle guiding the development of SCAFET was to create a feature detection and tracking technique that relies on minimal background information from climate datasets. The utilization of the Shape Index, a parameter bounded and scale independent, was well-suited to achieve this objective. While the curvedness parameter could offer supplementary insights into the identified features, instituting a threshold for curvedness, an unbounded variable, would counter the foundational concept of SCAFET. As evidenced in the manuscript, this supplementary filtering, which dilutes SCAFET's core principle, is not essential for feature identification.

   Nonetheless, we acknowledge that curvedness could still be advantageous in further filtering the identified shapes. Therefore, we have included an additional sentence in the paper to acknowledge this potential usefulness, encouraging new researchers to explore and pursue this avenue for enhancing feature detection with SCAFET.

4. The Hessian of a surface, $z(x, y)$, is a second-order term in the surface's local Taylor series expansion;its use here is only applicable if the first-order term (the gradient of $z(x, y)$) is zero. Indeed, even the cited reference that the authors rely upon, Koenderink and van Doorn (1992), contains the strong caution that the interpretation of shape from the Hessian matrix (emphasis theirs) "is only valid in representations where the magnitude of the gradient of z vanishes." This would seem to suggest that a preprocessing step to find critical points (where the gradient is zero) is necessary, but I did not see such a step mentioned.

   Yes. It is true that approximating the principal curvature as the eigenvalues of the Hessian of the surface $z(x, y)$ is true only if the point satisfies the condition $(\partial f/\partial x)^2 + (\partial f/\partial y)^2 = 0$. We have compared the SI from eigenvalues of Hessian to numerically calculated principal curvatures to find that,

   (a) In most of the real world applications, the difference in SI calculated from the eigenvalues of the Hessian are very close that from principal curvatures (see Figure 1, Figure 2, and Figure 3). In fact some popular implementations of Koenderink and Doorn 1992 uses the eigenvalues of the Hessian to calculate shape index (see https://scikit-image.org/docs/stable/api/skimage.feature.html#skimage.feature.shape_index)

   (b) Selecting a threshold of SI>0.375 inherently ensures that the identified regions are situated in the vicinity of points where the condition $(\partial f/\partial x)^2 + (\partial f/\partial y)^2 = 0$ holds true.

   To explicitly calculate the principal curvatures, we have followed Kurita and Boulanger 1992 to

first derive the Gaussian curvature (K) and mean curvature (H) as,

$$K = \frac{\partial f_{xx} \partial f_{yy} - (\partial f_{xy})^2}{(1 + f_x^2 + f_y^2)^2} \tag{1}$$

$$H = \frac{(1 + f_x^2) f_{yy} + (1 + f_y^2) f_{xx} - 2 f_x f_y f_{xy}}{2(1 + f_x^2 + f_y^2)^{1.5}} \tag{2}$$

Finally the principal curvatures are explicitly derived by solving the quadratic equation,

$$\kappa_{1,2} = H \pm \sqrt{H^2 - K} \tag{3}$$

Below, a comparison is presented regarding the values of SI obtained using both the eigenvalues of the Hessian and the principal curvatures from Equation 3. We see that in simple geometric fields, and climate data such as integrated vapor transport (used for identifying ARs) and relative vorticity (used for identifying cyclones), the values derived from both methodologies exhibit a remarkably close correspondence.

[Figure]

Figure 1: Comparison of the shape index (SI) calculate using the eigenvalues of the Hessian of the input field and the numerical method. (a) depicts a simple geometric field given by $5sin(.035x) - 2cos(.10y)$ used. Shading in (b) shows the SI calculated using the principal curvatures obtained using Equation 3. The contours in (b) display the SI derived from the eigenvalues of Hessian of (a). Note that the contours are drawn for the same values. (c) shows the difference between SI calculated using eigenvalues and principal curvature.

[Figure]

Figure 2: Same as in Figure 1 but for integrated vapor transport. This field is used for the detection of atmospheric rivers.

[Figure]

Figure 3: Same as in Figure 1 but for relative vorticity. This field is used for the detection of cyclones.

5. The discussion of the method's extension to 3D needs more detail. Given the horizontal-vertical splitting common to climate data sets, this is a nontrivial task. All of the shape index mathematics are formulated for 2D problems (with only 2 eigenvalues); for the 3D demonstration, the choice to use $k_1$ and $k_3$ and exclude $k_2$ seems significant. What is its physical interpretation? Perhaps this is justifiably outside the scope of the paper, but I wonder how would the definition of shape index change to account for all three eigenvalues? What are the analogous "shapes?"

We agree that an in-depth investigation into the physical interpretation of the shape index would provide significant insights. Unfortunately, given our current knowledge constraints and the manuscript's scope, we were unable to undertake this extensive analysis. Nonetheless, in order to illustrate the shape index's efficacy in identifying 3D structures embedded in a scalar field, and to maintain consistency with our approach of presenting simple examples, we have introduced a new appendix section. This section demonstrates the detection of a cylindrical volume within a three-dimensional scalar field. Notably, this particular problem bears substantial resemblance to identifying 3D jet cores (see Figure 4).

All prior studies involving the Shape Index (SI) have been developed for two-dimensional fields. We have extended its applicability to three-dimensional data by calculating the SI using all three possible combinations of eigenvalues. The decision to omit the smallest eigenvalue, denoted as $k_3$, is rooted in empirical observations. As discussed, SI can be computed as $SI(k_1, k_2)$, $SI(k_1, k_3)$, and $SI(k_2, k_3)$, with the condition $k_1 \geq k_2 \geq k_3$.

Empirical evidence indicates that for a convex curvature (regions exhibiting positive maxima), both $SI(k_1, k_2)$ and $SI(k_1, k_3)$ effectively capture the shape, while $SI(k_2, k_3)$ holds true in a trivial manner (refer to Figure 4). Conversely, a concave shape is accurately represented by both $SI(k_1, k_3)$ and $SI(k_2, k_3)$, whereas $SI(k_1, k_2)$ is trivially applicable.

Hence, the decision regarding identifying a local maximum ultimately comes down to choosing between $SI(k_1, k_2)$ and $SI(k_1, k_3)$ or utilizing both. Our analysis (see Supplementary Figure 4 and Figure 4) demonstrates that an SI based on $SI(k_1, k_2)$ enforces a stricter criterion, selecting a narrower region in proximity to the local maxima, while $SI(k_1, k_3)$ encompasses a slightly broader area. Similarly, for concave regions, SI based on $SI(k_2, k_3)$ can be chosen.

In essence, one can conclude that constructing the SI using the two largest eigenvalues yields the

most strict identification of a convex structure, while employing the two smallest eigenvalues yields the most rigorous detection of a concave structure.

These discussions and Figure 4 are now included in the new appendix and the relevant section in the manuscript.

[Figure]

Figure 4: The figures depict the various approaches for extracting a 3D cylinder from a scalar field. In (a), a simple scalar field is represented by $sin(3X) + cos(4Y) * cos(Z)$. Subsequently, in (b), the function is displayed following the application of the condition $SI(k_1, k_2) > 0.375$, while in (c), the function is presented with the condition $SI(k_1, k_3) > 0.375$ imposed. Finally, in (d), the selected regions within the field that fulfill the condition $SI(k_2, k_3) > 0.375$ are shown. The values enclosed in parentheses within the figure titles indicate the percentage of data that satisfies the respective applied condition.

6. The notation in equation (2) is confusing. It is customary for the numerator to contain the maximum order of each derivative; here, in all four terms the superscript 2 is missing. In the mixed terms, the symbol $\partial$ is missing between the x and y. The same comment applies to the indices.

Sorry about the typo. The mistakes has been rectified.

7. I prefer to see the functional dependence of newly introduced methods defined explicitly; for example in Equation (3), $SI(k_1, k_2)$ is more informative than simply SI, and $\phi_s$ (x, y) should match $\phi_p(x, y, ...)$ in equation (1).

That is a good convention. The equations will be modified accordingly.

8. What do the dots (. . .) represent in equation (1), and in Figure 1's definition of $\phi_s$ in the exponent?

The dots represent different dimensions of the data. The data could be one, two or n dimensional.

9. The EGU audience is interdisciplinary, and some may not be as familiar with the properties of the Hessian matrix and its eigenvectors. Adding their illustration to the dictionary of shapes in Figure 2 would be very helpful.

It had been attempted before. But it made the figure extra confusing. The eigenvectors are shown in the AR detection in figure 4.

10. It's not clear how distinct and/or subjective the boundaries between different SI regions are; for example, how different is a "Rut" with SI = -0.374 from a "Saddle Point" with SI=-0.375? What about other boundaries, e.g., Ridges and Saddles, Ridges and Caps/Domes? How does this affect the various features that are sought — how easily could a "cap" be misclassified as a "ridge," and how significant might that be to the results of a study?

Sensitivity test to shape index and other parameters are now included in the new appendix section. This addition aims to provide a more comprehensive understanding on how slight variations the smoothing scale $(\sigma)$, SI and other variables affect the detected AR properties.

11. A validation study comparing SCAFET to existing methods (such as TECA, which the authors mention) would be helpful — given the same criteria, do they detect the same tropical cyclones? If there are differences, what are the characteristics of the storms that appear in one but not the other. Similarly, given the emphasis of the work on large data sets produced by high resolution models, how does the computational performance of the proposed method compare with previous methods? Does the method achieve faster processing times? Both variety of studies were performed in [1], which should also be cited here.

A comprehensive examination of the similarities and differences in the detected characteristics between SCAFET and other methods extends beyond the scope of this study, as the method is developed for the detection of multiple features, not just one. However, we have added an additional supplementary section that presents a comparison of the identified atmospheric river (AR) attributes using SCAFET and alternative detection methods. Specifically we have compared the memory usage, computational demands, and processing time between SCAFET with alternative methods like Guan and Waliser 2015 and Ullrich et al. 2021.

We anticipate that the thorough comparison, along with case studies such as those carried out within the ARTMIP framework (https://www.cgd.ucar.edu/projects/artmip) by the broader community, will provide further valuable insights. Furthermore, we have referenced a new study in which SCAFET is employed for identifying changes in ARs in response to increased greenhouse gas concentrations.

12. The paragraph beginning on Line 35 suggests that defining thresholds for particular features is challenging and can vary between and even within individual models. The implied suggestion is that the presented method, SCAFET, would solve this problem; however, the remainder of the work relies on the same expert analysis (for example, Table 1) that the paper claims to avoid elsewhere. Similarly, Table 2 presents a set of well-defined criteria for tropical cyclones that contradicts the Line 35 paragraph.

We acknowledge the apparent contradiction between the content on line 35 and the subsequent portions of the text. We want to clarify that this discrepancy arises primarily due to a lack of specificity rather than a blatant error. The pivotal aspect in the feature detection process entails the initial identification of potential candidates or the segmentation of the input field. Parameters such as minimum intensity, length, area, and the like, assume a comparatively minor role in shaping the overall characteristics of the feature. For example, designating cyclones as possessing a "cap or dome" shape is the fundamental step, while the other parameters function as

filters that help isolate big*ger* or strong**er** cyclones. Consequently, these parameters in the tables need not be exceedingly precise or reliant on the specific dataset or grid employed. Importantly, they need not be reevaluated when applied to climate change studies.

This is also evident when you consider the different Atmospheric River detection algorithms discussed in Shields et al. 2018. Most detection algorithms in the catalog share a similar filtering process, diverging primarily in their approaches to the initial segmentation of the input field. The critical factor influencing the global distribution of ARs or their responses to global warming primarily hinges on this initial step Lora, Shields, and Rutz 2020; O'Brien et al. 2022.

Therefore, while SCAFET does utilize broad parameters for filtering the features identified in the initial step, the true crux of the matter lies in that initial stage. This general filtering approach is evident in various feature extraction methods. In response to your feedback and that of the other reviewer, we have revised the manuscript to articulate more specific claims and steer clear of sweeping statements.

13. Consider re-drawing Figure 7 with a white background for hard-copy readers.

    The figure has been redrawn. The modification in the main manuscript will be implemented once it receives approval from the editor.

**4  Point-by-point Response to Specific Comments**

1. Figure 1: "Hessian" should be capitalized.

   The suggestion has been accepted and implemented all throughout the manuscript and supplementary section.

2. There are numerous grammatical and typesetting errors. An incomplete list includes:

   (a) The inverse tangent function in equation (3) should not be italicized; so should sin and cos in the Figure 2 caption.

   (b) Period missing after equation (3).

   (c) The symbols n and n + 1 in Section 2.3 should be italicized.

   (d) "circ" in Lines 202 and 236 should be $^\circ$.

   (e) km in lines 203, 204 should not be italicized

   (f) Please use either $tan^{-1}$ or $arctan$, not both.

   All of the suggestions you provided have been successfully incorporated. The manuscript has been subjected to additional refinement to rectify any lingering grammatical errors. Your meticulous review and commitment to detail are greatly valued and appreciated.

**References**

Guan, Bin and Duane E. Waliser (2015). "Detection of atmospheric rivers: Evaluation and application of an algorithm for global studies". In: *Journal of Geophysical Research: Atmospheres* 120.24, pp. 12514–12535. DOI: 10.1002/2015jd024257. URL: https://doi.org/10.1002%2F2015jd024257.

Koenderink, Jan J and Andrea J van Doorn (1992). "Surface shape and curvature scales". In: *Image and Vision Computing* 10.8, pp. 557–564. DOI: 10.1016/0262-8856(92)90076-f. URL: https://doi.org/10.1016%2F0262-8856%2892%2990076-f.

Kurita, Takio and Pierre Boulanger (1992). "Computation of Surface Curvature from Range Images Using Geometrically Intrinsic Weights". In: *Proceedings of IAPR Workshop on Machine Vision Applications*. Minato-ku, Tokyo, Japan: International Association for Pattern Recognition, pp. 389–392. URL: https://www.mva-org.jp/Proceedings/CommemorativeDVD/1992/toc1992.html.

Lora, J. M., C. A. Shields, and J. J. Rutz (2020). "Consensus and Disagreement in Atmospheric River Detection: ARTMIP Global Catalogues". In: *Geophysical Research Letters* 47.20. DOI: 10.1029/2020gl089302. URL: https://doi.org/10.1029%2F2020gl089302.

O'Brien, T. A. et al. (2022). "Increases in Future AR Count and Size: Overview of the ARTMIP Tier 2 CMIP5/6 Experiment". In: *Journal of Geophysical Research: Atmospheres* 127.6. DOI: 10.1029/2021jd036013. URL: https://doi.org/10.1029%2F2021jd036013.

Shields, Christine A. et al. (2018). "Atmospheric River Tracking Method Intercomparison Project (ARTMIP): project goals and experimental design". In: *Geoscientific Model Development* 11.6, pp. 2455–2474. DOI: 10.5194/gmd-11-2455-2018. URL: https://doi.org/10.5194%2Fgmd-11-2455-2018.

Ullrich, Paul A. et al. (2021). "TempestExtremes v2.1: A Community Framework for Feature Detection, Tracking and Analysis in Large Datasets". In: *Geoscientific Model Development*. DOI: 10.5194/gmd-2020-303. URL: https://doi.org/10.5194%2Fgmd-2020-303.